# Numerical Simulations of Potential Contribution of the Proposed Huangpu Gate to Flood Control in the Taihu Lake Basin of China

Zhang Hanghui[1,2], Liu Shuguang[1], Ye Jianchun[2], Pat J.-F. Yeh[3]

[1] Department of Hydraulic Engineering, Tongji University, 200092, Shanghai
[2] Taihu Basin Authority of Ministry of Water Resources of P.R.China, 200434, Shanghai
[3] Department of Civil and Environmental Engineering, National University of Singapore, 117576,Singapore

*Correspondence to:* Liu Shuguang (liusgliu@tongji.edu.com)

## Abstract

The Taihu Lake basin (36895 km$^2$), one of the most developed regions in China located in the hinterland of the Yangtze River Delta, has experienced increasing flood risk. The largest flood in history occurred in 1999 with a return period estimate of 200 years, much higher than the current capacity of flood defense with the design return period of 50 years. Due to its flat saucer-like terrain, the capacity of flood control system in this basin depends on flood control infrastructures and peripheral tidal conditions. The Huangpu River, an important river of the basin connecting Taihu Lake upstream and Yangtze River Estuaries downstream, drains two-fifth of the entire basin. Since the water level in Huangpu River is significantly affected by the high tide conditions in estuaries, constructing an estuary gate is considered as the most effective solution for flood mitigation. The main objective of this paper is to assess the potential contribution of the proposed Huangpu gate to the flood control capacity of the basin. To achieve this goal, five different scenarios of flooding and the associated gate operations are considered by using numerical model simulations. Results of quantitative analyses show that the Huangpu gate is effective to evacuate floodwaters. It can help to reduce both the peak values and the duration of high water levels in the Taihu Lake to benefit the surrounding areas along the Taipu Canal and Huangpu River for more than 100 km2. The contribution of the gate to the flood control capacity is closely associated with its operation modes and duration. For the maximum potential contribution of the gate, the net outflow at the proposed site is increased by 52%. The daily peak level is decreased by a maximum of 0.12m in the Taihu Lake, by a maximum of 0.26-0.37 m and 0.46-0.60m in the Taipu Canal and Huangpu River, respectively, and by 0.05-0.39m in surrounding areas along the two rivers depending on local topography. It is concluded that the proposed Huangpu gate can reduce flood risks in the Taihu Lake basin and the surrounding areas along the Taipu Canal and Huangpu River significantly, which is of great benefits to the flood management in the basin and the Yangtze River Delta.

**Keywords:** Flood control; Huangpu Gate; Taihu Lake Basin; Numerical analysis

## 1 Introduction

The Huangpu River, located in the downstream part of the Taihu Lake basin, is the main shipping and drainage route to the port city Shanghai in China. It flows through the urban core of Shanghai city, which is evaluated as one of the most vulnerable metropolises to extreme flooding in the world (Balica et al., 2012). Wang et al. (2012) predicted that half of Shanghai will be flooded and 46% of seawalls and levees will be overtopped in 2100, causing serious urban flooding. Typhoon is one of main natural factors to trigger flood disaster in this area. When typhoon comes, the storm surges caused will be driven into the Yangtze River estuary to further increase storm tide levels due to the shallow waters and confined dimensions within the estuary (Nai et al., 2004). When this coincides with the astronomical high tides, the storm tide traveling into the Huangpu River can rapidly raise water levels in river and possibly cause inundation of the urban areas of Shanghai. It has been reported that along with global climate change, the frequency and intensity of typhoons have increased substantially (Qin et al., 2005).

Taihu Lake is located about 80 kilometers away west of the Shanghai city center (Figure 1). The Huangpu River is the major river draining floodwaters of both Shanghai city and Taihu Lake basin. After the completion of eleven key projects for the integrated water resources management in the basin, the discharge from the upper reach of Huangpu River increased, resulting in a considerable water level rise in the Huangpu River (Zhou et al., 2016). The river embankments, a traditional flood defense infrastructure, was built along the Huangpu River in 1950s. Its flood control capacity, however, has been decreased by increasing storm surges and extreme tides, man-made changes in the estuary, land subsidence, and aging infrastructures. Currently, the river embankments need to be periodically raised to withstand the increasing water levels.

The designed return period of the Huangpu River embankment approved in 1985 is 1000 years. The historical highest water levels recorded was recorded during the 11th typhoon in 1997. At the Huangpu Park observational station near the Shanghai city core, the water level reached the historical height of 5.72 m (0.5 m higher than the second largest record in 1981) and only 0.14 m lower than the design water level at this location (Nai et al., 2004). Based on the revised hydrological analyses which extended the water level series from 1912-1983 to 1912-2002, the embankment height in its original design corresponds to less than 200-year return period due to the newly recorded high tide in 1997 (Shao, 1999; Yao, 2001; Lu, 2008). In 2004, the standard of 1000-year recurrence level was found to be degraded to the 100-year level mainly due to sea-level rise and land subsidence (Tang et al. 2014), indicating that the flood protection capacity was reduced. To enhance the flood protection capacity of the Shanghai city, the height of embankments has to be raised to meet the standard of 1000-year return period. However, the continuous increase of the height will not only require huge economic cost but also affect urban landscape and water environment, with another potential risk being that the extreme dam-break flooding will be more devastating. In addition, the reliability of the reinforced embankment structure is in question because of its aging foundation built around 1950s (Zhou et al., 2016).

A combination of the flood defense walls and estuary barriers has been proposed as an alternative measure to against the reduced capability of flood control in the low-lying regions in England, Netherland, Germany, among other countries (Xiao, 2017; Jin, 2016). The Thames barrier

in UK, for instance, has been operated for more than 30 years with a significant flood control capacity for protecting the large cities upstream. It can effectively mitigate flood risks caused by discharge from upstream areas and high tides caused by storm surges (EA, 2012). Xiao (2017) reported that an area of 125 km$^2$ in London can be protected against the high water level of 1000-year return period when the Thames barrier is completed. After the completion of the Delta Storm Surge Barriers project in the Dutch delta, the protection standard was increased from once in the return period of 1250 years to that of 4000 years, protecting one third of the areas of Netherlands as well as 4.5-million people. The Aames tidal gate in Germany has raised the level of protection against storm surge from the North Sea up to 3.7 meter above the mean sea level. Inspired by the above international experiences of flood protection, since 1998 the Municipal Government of Shanghai city has been investigating the feasibility to protect this area with a storm surge barrier at the mouth of the Huangpu River.

As the Huangpu River runs through one of the most important metropolitan areas in China, numerous studies since 1990s have demonstrated the significance of constructing an estuary gate to enhance the safety of the Shanghai city (Chen, 2001; Shao, 1999; Shao and Yao, 1999). Chen (2001) and Shao (1999) carried out comparative studies based on the experiences of the well-known Thames Barrier in UK and the Delta Storm Surge Barriers in Netherlands. Jin (2016) conducted an in-depth analysis of typical large tidal gates built globally on the aspects of planning and design, investment and construction, and operation and maintenance. Chen (2002(a)) estimated the economic benefits in terms of the protected areas by the proposed tidal gate at the estuary of the Huangpu River.

Most afore-mentioned studies on the importance and necessity of constructing the estuary gate are based on comparative and qualitative analyses only. Although some previous research provided quantitative estimation for the potential benefits of the proposed gate (Chen, 2002(b); Cui, 2012), the majority of them only considered the role of the gate in blocking the tide intrusion for the local estuary areas of the Huangpu River. Few research have provided a holistic evaluation on the potential contribution of the proposed gate to flood control for the entire Taihu Lake basin, in particular the synergistic effects for the upstream areas of the basin due to gate construction. As the Huangpu River connects the Taihu Lake with the Yangtze River estuary to drain floodwater from both local and lake upstream areas, the investigation of potential contributions of the proposed gate to flood control for the entire basin is of great engineering significance, and this is the main thrust of this study. To achieve this goal, various scenarios of the monsoon-induced floods are analyzed in this study and their impacts are quantified through numerical modelling simulations.

## 2 Study Area

The Taihu Lake Basin, located in the hinterland of the Yangtze River delta, is one of the most developed areas in China. The Taihu Lake is located in the center of the basin, surrounded by the Yangtze River on the north, the Hangzhou Bay on the south, and the East China Sea on the east, as shown in Fig. 1. This basin is not a sizable basin with the total area of 36,895 km$^2$, only 0.4% of the national total (Hu and

Wang, 2009). However, the Gross Domestic Product (GDP) is up to RMB 6.69 trillion by the end of 2015, accounting for about 10% of the national total, and the regional per capita GDP being more than 2.5 times of the national average. This region is of great significance for the social and economic development of China. However, the extensive urban development has contributed to the risk of increasing flood magnitude and frequency over this region.

The Taihu Lake Basin is characteristic of a complex hydro-system that includes interlaced rivers, dense water nets, and dotted depression lakes of different sizes (Qin, 2008). The water network and drainage system in the basin possesses the following unique properties: (1) it has a saucer-like landform, and the elevation of more than half of the floodplains is lower than the water level of flood control; (2) it is a typical river plain region with a high river net density of 3.2 km/km$^2$ and the total river length of about 120,000 km; (3) the surface gradient is about 1/100,000 - 1/200,000, and the river flow velocity is only 0.3 - 0.5m/s in flood seasons; (4) the daily drainage time of the peripheral outlets in the basin is about 13 - 14 hours due to the semi-diurnal tides. Overall, the capacity of flood control system in the basin is dependent to a large extent on the flood defense infrastructure and the peripheral tidal conditions. Based on the characteristics of topography and water networks, the basin is divided into eight sub-areas, namely the Huxi, Wuchengxiyu, Zhexi, Taihu Lake, Yangchengdianmao, Hangjiahu, Puxi, and Pudong (as shown in Fig.1). The irrigation systems were built which control the water exchange among these sub-areas. The unique saucer-like topography of the Taihu Lake dictates that the water storage is easy to accumulate but difficult to drain, hence renders the surroundings flood-prone areas (Gao et al., 2005).

The Taihu Lake basin lies in a subtropical climate zone characterized by mild temperatures, high humidity and abundant rainfall (long-term average 1177mm/year). The basin is prone to both monsoon-induced and typhoon-induced floods. Major flood disasters, with the inundation areas greater than 3000 km$^2$, have occurred more than ten times during the twentieth century (Yu et al. 2000). The largest flood disaster occurred in 1999 resulted in damages with direct economic loss of 16 billion USD (Wang et al., 2011). There are 239 typhoons hitting the basin during the 1949-2013 period, on average about 3 to 4 per year (Ye and Zhang, 2015). According to the recent assessment report (AR5) compiled by the Intergovernmental Panel on Climate Change (IPCC, 2013) where the flood control of coastal systems and low-lying areas is addressed, the Yangtze River delta is identified as one of the highly vulnerable coastal deltas in the world.

Generally, the basin is characterized by the monsoonal climate with the period concentrated in summer (from June to July), lasting several weeks or even months. Consequently, the broad-scale rainfall events occur frequently with an excessive magnitude and a long duration, contributing to the basin-wide floods. The largest flood in history occurred in 1999, and the total rainfall in a 43-day monsoon period reached 670 mm, three times more than the long-term (1954-2010) average during the same period of the year. The return period of 1999 flood event is estimated as 200-year (Evans and Cheng, 2010), much larger than both the current 50-year design return period of the flood control capacity of the basin. Mean totals of the 7-day, 15-day, 30-day, 45-day, 60-day and 90-day accumulated rainfall in 1999 all exceeded the historical records (Wu, 2000). During this flood, the high water level in Taihu Lake set a new record of 5.08 m, exceeding the design water level of 50-year return period by 0.43 m.

In the basin, there are numerous tidal channels linking the lakes and the coast (bay, estuary), and most outlets are controlled by the floodgates subject to tidal locking. The Huangpu River meandering through the downtown area of Shanghai City connects the westward-located Taihu Lake with the Yangtze River estuary in the North East, as shown in Fig. 1. The Huangpu River is 113 kilometers long, with a depth of 5-15 meters and a width of 300-500 m (800 m at the estuary), formed by the convergence of Xietang River originated from the Taihu Lake and Area Yangchengdianmao, Yuanxiejing Creek and Maogang Creek from originated from Area Hangjiahu. The river flows through the downtown Shanghai before finally injecting into the Yangtze River at the estuary mouth of Wusong, as the only one without an estuary gate in the basin. The tidal effect complicates the flow patterns of the Huangpu River, and helps to keep the floodwater in the river. Generally, the river can naturally drain floodwaters only for 13-14 hours per day. The Huangpu River experiences two high tides and two low tides each day (semi-diurnal tides), receiving about 40.9 billion $m^3$ of tidal water from the Yangtze River (Zhang, 1997). The total tidal influx of the Huangpu River is about 47.47 million $m^3$/year, and the total inflow from its upstream area is about 10.02 billion $m^3$/year. Its sediment concentration upstream is of 0.049 kg/$m^3$, and that is of 0.213 kg/$m^3$ downstream (Yan, 1992). The problem caused by sediment and siltation is not serious in this river because the inflow from the upstream area is far more than the tidal water from the estuary.

## 3 Methodology

### 3.1 Description of five scenarios

It is instructive to investigate the potential contribution of the proposed Huangpu gate to the flood control of the Taihu Lake basin, which is still in the preliminary demonstration-of-benefit stage currently. The main research strategy used in this study is the scenario analysis based on numerical model simulations.

In total five different scenarios are considered in this study as summarized in Table 1. Among them, the first scenario considers the case without gate construction, and all the remaining four scenarios consider the cases with gate construction, but each with different operational rules. The scenario "base A" is used as the basis for comparison with other scenarios (Table 1). It represents the case that the estuary gate is not constructed at the Huangpu River mouth. The scenarios "A1" and "A2" are designed for the quantitative analysis of the potential benefits due to gate operation in draining floodwaters from the Taihu Lake and its upstream areas. The scenario "A3" is designed to analyze the contribution (function) of gate construction to block tide intrusion. The last scenario, "A4", is the case for analyzing the potential maximum benefits to flood control of the basin. The numerical simulations for the five different scenarios are all based on the conditions during the 1999 flood event, which is the largest flood in history for the study area.

For the scenario "base A", the estuary gate is not constructed at the outlet of the Huangpu River. Thus, the water in the Huangpu River and Yangtze River estuary can exchange naturally. For the scenario "A1", the proposed gate will be operated in the rising stage of the lake levels according to weather forecast. In the model simulation of the 1999 flood, it began to operate seven days in advance before the lake level reached its peak value.

For the scenario "A2", the proposed gate will be operated when the large basin-wide floods occur with the lake level higher than 4.50 m, meaning a severe and urgent flooding situation in the Taihu Lake which requires the acceleration of floodwater drainage of the major downstream rivers including the Huangpu River.

For the scenario "A3", a portion of the tidal water intrusion would be blocked by the gate, and the gate will remain open to prevent tide intrusion until the tide rises to a threshold (defined in this study as 4.0m). That is, the gate will not be closed for blocking tide intrusion every day; instead, it will only be closed under the situation when the high water exceeds the tide threshold and is forecasted to continue rise.

For the last scenario "A4", the gate would prevent all tidal water intrusion during the flooding period, representing a hypothetical extreme case since it is not practical in implementation owing to the difficulty in frequent operation (e.g., to close twice every day) of such a huge gate with a width of about 400–500 m. This scenario is merely a case for analyzing the potential maximum benefits to flood control of the basin. Indeed, it is not necessary to block all tidal water intrusion. Under such condition, it is also likely

to produce negative impacts on both waterway transportation and water environment system.

Considering the time needed for making policy decision and gate construction, it is highly likely that the Huangpu gate will not be completed and start to operate after 2025. For this reason, the proposed flood control projects in the plan designed by MWR (2008) will also be incorporated as the scenarios of flood defense in this study. The model parameters of the numerical simulations in this study are specified

as the same as those used in the design plan by MWR (2008). Hu (2006) proposed the Anchorage Ground located at the mouth of Huangpu River (as shown in Fig. 1) as the best site for the gate construction since the negative impacts to the shipping and navigation are the least due to its location. Cui (2012) and Lu (2008) proposed the same location for gate construction. Similarly for the numerical model simulations in this study, the estuary gate will also be simulated at the Anchorage Ground, the place name in the

unpublished master's thesis with English abstract (Hu, 2006), which is about 5-6 km from the Huangpu River mouth.

## 3.2 Model description

The HOHY model developed by the Hohai University in China is used in this study. This model has been tested for numerous regional applications since 1970s, and was applied to study the Taihu Lake basin in

1997. It is one of the main products of a three-year water study at the Taihu Lake basin, supported by the World Bank and jointly undertaken by the Hohai University and the Delft Hydraulics in Netherlands. The HOHY model can simulate the cycle of floodwaters well. Meanwhile, the model can provide a broad-scale simulation of the flood control system in the Taihu Lake basin. It can simulate not only complex hydro-systems with numerous interlaced rivers and lakes, complicated relationships between

river nets, hilly topography and tidal boundaries, also the complex operational rules of control structures, such as sluices, pumps and siphons. This model has been utilized in a variety of past studies, such as the preliminary demonstration-of-benefit stage of water works in the Taihu Lake basin. In particular, the model has been successfully applied in the flood control planning of the Taihu Lake basin as approved by WMR (MWR, 2008).

The model is composed of two parts: a hydrological part for simulating runoff generation and routing and a hydraulic part for simulating channel flows. Each of them can run independently. The schematization of the model is shown in Fig. 2. More details of the model can be found in Cheng et al. (2006) and Jin (2009).

5    Runoff is generated when precipitation exceeds the infiltration, interception and depression storage. The basin land use is classified into four types: water surface, paddy field, non-irrigated farmland and constructed land. Each of them employs different parameterizations to calculate runoff generation. Runoff is then routed according to basin topography. In hilly areas, the instantaneous unit hydrograph method is used, considering the store and drainage processes of reservoirs and large ponds. In plain areas, 10    the method of runoff curve number is used for each computed area.

After the runoff from the hilly and plain areas flows into river networks, the hydraulic method is applied for simulation of river flow. Only those lakes with larger water surface are considered as possessing the function of storing floodwater, while the others are considered as intersections like the links among rivers. The operation of water-engineering works such as the simulations of gates, pumping 15    stations and siphons will be simulated in the model. The Saint Venant Equations are used as the governing equations for the 1-D unsteady open channel flow, including the continuity equation (1) and momentum equation (2) as follows

$$\frac{\partial Q}{\partial x} + \frac{\partial A}{\partial t} = q_L \tag{1}$$

$$\frac{\partial Q}{\partial t} + \frac{\partial}{\partial x}(\alpha \frac{Q^2}{A}) + gA \frac{\partial Z}{\partial x} + gA \frac{n^2 |Q| Q}{R^{1.333}} = q_L v_x \tag{2}$$

20    where $x$ is the distance along each channel; t is the time; $A$ is the cross section; $Q$ is the flow rate; $Z$ is the water level; $\alpha$ is the momentum correction coefficient; $R$ is the hydraulic radius; $q_L$ is the lateral inflow per unit length of channel; Vx is the velocity of the lateral inflow in the x-direction; and g is the gravity acceleration.

The model calibration data are from two consecutive years 1984 to 1985, and the validation data 25    are from 1995 and 1996 (Liang et.al, 1993). The model is tested for the 1999 flood events by detailed comparisons between model simulation and observational data simulated by Ou and Wu (2001). Fig.3 compares water level differences between model simulations and observations in 1999 flood at eight representative stations in the basin (see Fig.1 for the locations of eight stations). Fig.4 shows the difference in river discharges between observations and simulations at the Taipu Gate and Wangting 30    Siphon (as shown in Fig.1), respectively. The comparison of simulated water levels and discharges with observations demonstrates that the HOHY model simulations are of sufficiently high accuracy for the following analyses.

Among the five scenarios considered, the operational rules in the scenario A3 are the most complex to simulate since the following different operational rules of the gate will be applied for the flood tide 35    and ebb tide respectively. If the high water in the flood tide is higher than the tide threshold, the gate will be closed. Once the gate is closed, it will not be re-opened until it has the natural water-expelling ability to drain floodwater in the ebb tide (until the tide level falls to be lower than the water level in the upstream of the gate). Hence, the HOHY model needs to modify in order to enhance its capability for this purpose.

The model modification focuses on the part of flood routing related to the algorithms of unsteady open channel flow, and the control rules of gates related to the tidal conditions. The main program was improved by adding a function to judge the stage of tide before running the gates (i.e. in the flood or ebb tide), which makes the specification of the control rules of gates more flexible. The original program is modified based on the flowchart given in Fig. 2.

The modified model is tested by using a simple case of which the tide threshold is assumed to be 4.0 m. The simulation results are presented in Fig.5 and Fig.6. The first figure describes the case when the gate remains open since the high water in this tidal period is always lower than 4.0 m. The second figure, as a comparison, is the case that the gate should be closed when the rising tide is higher than 4.0 m. The gate will re-open to drain floodwater when the gate has the natural water-expelling ability to evacuate floodwater in the ebb tide.

The model results including the gate discharge, the tide water level at the estuary, and the difference of water levels between the upstream and downstream of the gate, show the reasonable relationships of the operational rules of the gate. The results in Fig.5 demonstrate that the discharge at the gate resembles a sinusoidal curve as affected by the tidal boundary. It is likely that the gate needs not to be closed since the high water in this tidal period is less than the tide threshold of 4.0 m. Fig.6 is another case of the gate operational rules of which the high water is about 4.70 m. At 2:30 am, 15/August/1999, the tide level at the river outlet in the flood tide slightly exceeded the tide threshold of 4.0 m, and the gate has to be closed. It was not re-opened in the ebb tide until 8:15 am, 15/August/1999 when water level in the upstream is higher than that in the downstream near the proposed gate, meaning that the gate has the natural water-expelling ability to drain floodwater at this moment (see the red bars in Fig.6). Overall, the modified model has demonstrated the ability to simulate the complex operational rules of the proposed gate.

## 4 Result analysis and discussion

Based on the topography and water systems of the study region, the Huangpu River receives floodwaters from the Taihu Lake and the surrounding areas draining into the Taipu Canal and Huangpu River, in particular those low-lying areas in the southern part of the Yangchengdianmao catchment, the northern part of the Hangjiahu catchment, and the western part of the Puxi catchment (Fig. 7). Therefore, the potential contributions of the proposed Huangpu gate to the flood control capacity will be analyzed in the following section with respect to three target regions: (1) the Taihu Lake, (2) the surrounding areas, and (3) the Taipu Canal and the Huangpu River.

### 4.1 Potential contribution to flood control of the Taihu Lake

Table 2 summarizes the peak lake water levels and the duration (the number of days) during the June-August period in 1999 when various control water levels were exceeded under five different scenarios considered in this study. Fig. 8 plots the simulated water levels at the Taihu Lake corresponding to five different scenarios during the 1999 flood event from June to August. As seen, the lake levels in the scenarios A1, A2, A3 and A4 are all lower than that in the base A scenario. Similarly, the durations when

the water level is higher than certain control levels were also reduced. Compared with the maximum daily water level of 5.03m (occurred in early July) in the base A scenario, the maximum water levels in other scenarios were decreased by 0.04 m, 0.01 m, 0.03 m and 0.12 m, respectively, for the scenarios A1, A2, A3 and A4. Thus, these four scenarios contribute to the flood control capacity of the Taihu Lake and its adjoining low-lying areas to the west.

It should be noted that the differences in the design water levels corresponding to different return periods are not significant for such typical of shallow lakes located in the low-lying plains. For instance, the design water level of 100-year return period is 4.80 m, 0.15 m higher than that of 50-year return period. For this reason, the decrease in the peak lake level by 0.04 m in Scenario A1 as well as by 0.12 m in Scenario A4 is significant for the flood control of the lake. Additionally, the western adjoining floodplains also benefit from the proposed gate construction. Due to the relatively lower flood control capacity of the western adjoining areas, those regions are likely to be inundated when the sluices cannot yet control the water intrusion from the lake to the adjoining areas once if the lake level is too high. The flooding in the western adjoining areas will be even worse once if the lake breaches the dike.

From the viewpoint of flood control of the lake, it can be concluded that the Huangpu River with an estuary gate is more effective than the natural river without a gate. The extent of gate contribution to flood control depends largely on the total operation time of the gate. The longer the gate is operated, the more tidal water will intrude into the Huangpu River estuary. The gate operation will prevent and reduce the amount of tidal water from entering the upper estuaries that are already with high water levels caused by high river flows. Clearly, the floodwaters in the lake will be drained slowly by the Taihu Canal and the Huangpu River, which have high water levels due to tide intrusion. Overall, the scenario A1 is a good example to examine the potential contribution of the proposed gate. In the simulation of the 1999 flood event, the Huangpu gate is more effective to reduce flood risk in the lake by operating the estuary gate in advance. Even with the case of operating the gate by a relatively short period of time, such as one week as in this study, the contribution to reduce the peaks and slow down the rising rate of lake levels are rather significant.

## 4.2 Potential contribution to flood control of the surrounding areas

Floodwaters from the following surrounding areas are also drained into the Huangpu River: (1) Yangchengdianmao, (2) Hangjiahu, (3) Puxi, and (4) Pudong catchment, as shown in Fig. 7. Therefore, the safety of these four catchment areas against flooding is also closely linked to the capacity of the Huangpu River. Table 3 lists the peak water levels at the four representative stations (S1 to S4, shown as the orange circles in Fig. 1. each for one of the above four surrounding areas respectively). Fig. 9 plots the simulated daily water levels during the 1999 flood event at these four stations, from which the similar trend as the water level in the Taihu Lake (Fig. 8) can be observed. The scenario A4 represents the maximum potential contribution of the proposed gate, i.e., the maximum decrease in the daily peak level at the four stations in the surrounding areas is 0.32 m, 0.19 m, 0.39 m and 0.05 m, respectively. In contrast, the improvement in the flood control capacity at station 4 at the Pudong catchment is the smallest among four stations due to its unique terrain. The local floodwater in the Pudong catchment draining to the East China Sea has the priority over that draining to the Huangpu River due to its natural water-expelling

ability. Generally, the flood capacity of station 4 does not depend much on the drainage capacity of Huangpu River as the other three stations.

In contrast, for the scenario A1 in which the gate is operated in advance, the gate can play a notable role in reducing the peak water levels by the amount between 0.15m and 0.35m except for the station 4, while for the scenario A2, the gate only decreases the peak water levels by 0.07-0.15 m. However, the scenario A2 has more advantages in speeding up the drainage rate of floodwaters at the recession stage and shortening the waterlogging time.

## 4.3 Potential contribution to flood control of the Taipu Canal and Huangpu River

Fig.10 plots the simulated daily water levels in the Taipu Canal and Huangpu River at the seven cross-sections (as marked by the purple rectangular in Fig. 1). The daily water levels at the Taipu Canal and Huangpu River decrease to various extents when the gate is in operation. The scenario A4 represents the potential maximum contribution of the gate due to the complete prevention of tidal water intrusion. In this scenario, the maximum reduction of peak water level is 0.26-0.37m for the Taipu Canal is 0.26 - 0.37 m, and 0.46-0.60m for the Huangpu River.

The Huangpu River benefits more from the proposed gate construction than the Taipu Canal because the latter is located relatively farther away from the gate. The potential contribution of the gate can be attributed to the reduction of tidal water intrusion during the flooding period. Generally, the tidal intrusion is mainly concentrated on the lower reach of the Huangpu River, although the intrusion can propagate upward as far as more than 100 km from the estuary. The water level will rise in the upstream reach of Huangpu River when the gate is closed, and then the discharge rate will increase when the gate re-opens again due to the relatively large difference in water levels between the upstream and downstream side near the gate. Therefore, the gate can decrease the water levels of Huangpu River more markedly than that of Taipu Canal.

In the scenario A1, the gate is operated in advance during the rising stage of the lake level, and the peak flood level in the Taipu Canal and Huangpu River can be decreased considerably due to the enlargement of drainage capacity of the Huangpu River. In the scenario A2, the gate is operated when the lake level is higher than 4.5 m, and its contribution to the peak water levels is less than the scenario A1, while the draining rate in the recession stage is faster. If the gate is operated by blocking the high tide during the flood period (Scenario A3), the peak water levels at the Taipu Canal and Huangpu River are decreased during the spring tides. This conclusion is completely consistent with those discussed in the previous sections on the contribution of the proposed gate to the flood control of the lake and the surrounding areas.

## 4.4 Analyses of the inflow and outflows in the Huangpu River

Table 4 describes the inflow volumes from the upstream tributaries to the Huangpu River during the flood period. In addition to the Taipu Canal, there are many upstream tributaries originating from the northwest and southwest upstream areas of the Huangpu River (Fig. 7). In the scenario A4, the inflow volume from the southwest tributaries into the Huangpu River is up to 3.25 billion m$^3$, more than twice of that in the scenario base A (1.50 billion m$^3$). The inflow from the northwest upstream areas in the

scenario A4 is about 1.05 billion m$^3$, increases by 78% in comparison to that in the scenario base A (0.59 billion m$^3$). The inflow volume from the Taipu Canal is about 5.0 billion m$^3$, only increased by 27.2% compared to that in the scenario base A (3.93 billion m$^3$). In terms of the major inflows into the Huangpu River, the inflow volumes from the southwest and northwest upstream areas increase significantly in comparison to that from the Taipu Canal, suggesting that the Huangpu River plays a dominant role for these two upstream subareas.

Table 5 describes the tide intrusion and outflow volume at the site of the proposed gate during the flooding period. The proposed gate helps to improve the drainage efficiency of Huangpu River by preventing the river from tidal water intrusion. Compared to the scenario base A, the net outflow volumes at the gate site during the entire flooding period under other four scenarios are increased by 4% (scenario A1), 8% (scenario A2), 22% (scenario A3) and 52% (scenario A4), respectively. Fig.11 shows the comparison of simulated river discharges at the site of the proposed gate between the scenarios base A and A1 from June 27 to July 3 in 1999. The difference in river discharge between these two scenarios clearly reflects the difference in the drainage efficiency of the Huangpu River. Although the river discharge in the scenario A1 is only increased by 4% for the entire flooding period (increased from 7.20 to 7.46 billion m$^3$). It should be noted that the influence on the flood control during the gate operation period is more significant because the net outflow volume is nearly doubled by changing the bi-directional flow to the unidirectional flow.

## 5 Conclusions

Compared to a natural river channel, an estuary gate can prevent tidal water from intrusion into the upstream estuaries. This study shows that the construction of an estuary gate at the Huangpu River is an effective measure for evacuating floodwaters and reducing peak levels in the upstream reaches of the river. The potential contributions of the proposed gate are closely associated with its operation duration. Regarding the maximum potential contribution, the net outflow at the site of the proposed gate is increased by 52% for the entire flooding period in 1999, and hence the efficiency of the drainage capacity from the Taihu Lake to the Yangtze River estuary is significantly improved.

Constructing the proposed gate will benefit the Taihu Lake, the surrounding areas, and the two major river channels, namely the Taipu Canal and the Huangpu River. The inflow volume from the upstream tributaries into the Huangpu River is increased by 27% (in the Taipu Canal), 78% (in the north part of Hangjiahu catchment) and 117% (in the south part of the Yangchengdianmao catchment), respectively. Meanwhile, the daily peak level is decreased by a maximum of 0.12 m in the Taihu Lake, by 0.05-0.39m in the surrounding upstream areas depending on local topography, and by 0.26-0.37m and 0.46-0.60m, respectively, in the Taipu Canal and Huangpu River.

The potential contribution of the gate depends on the operating time of the gate. For the scenario A1, it is beneficial to decrease the peak flood level and slow down the water level rise during the rising stage. For the scenario A2, it is helpful to speed up the drainage rate during the recession stage, to reduce the duration of high water level, and to decrease the flood risk of the lake and its adjoining upstream

areas. For the scenario A3, it appears that the flood control is more effective during the spring tides. Generally, the contribution of the gate is more notable in August than in other months.

Overall, it is significant and effective to build an estuary gate at the outlet of the Huangpu River to improve the capacity of flood control against the basin-wide large floods. The implementation of the gate needs further investigation, including the feasibility assessment on the economics, environment and navigation. When the operation rules of the proposed gate are formulated, much attention should be paid to the navigation in the river so as to mitigate the adverse influences on the shipping.

## Acknowledgements

This study was sponsored by the Chinese National Science & Technology Pillar Program (No. 2014BAL05B02) and the program of the National Natural Science Foundation of China (No. 41672230).

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

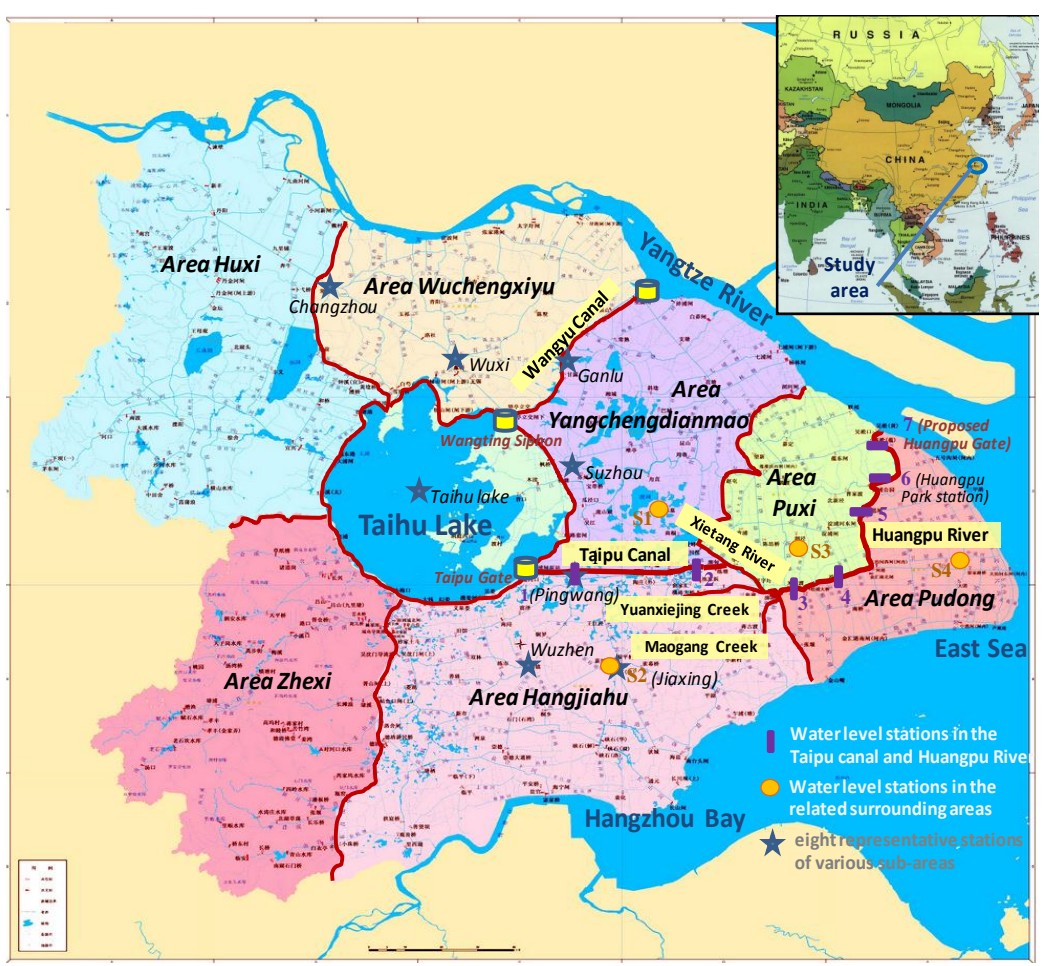

Figure 1: Location map of the Taihu Lake basin in the Eastern China.

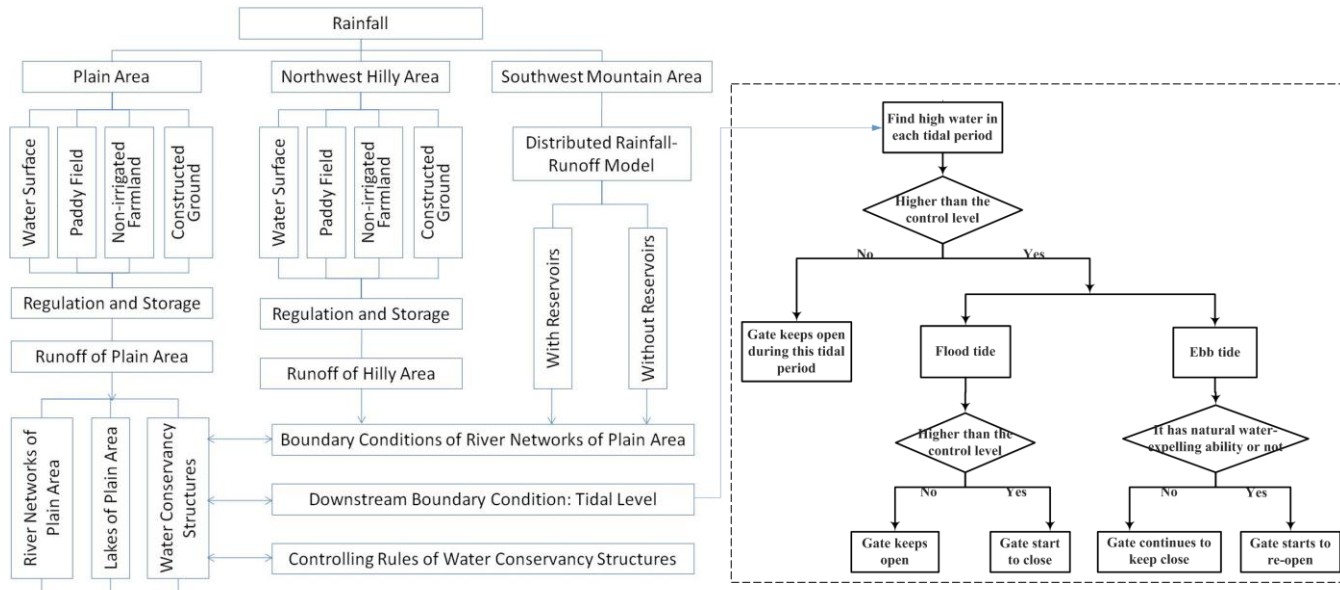

Figure 2: Schematization of the extended HOHY model (Modified from Jin et al., 2008)

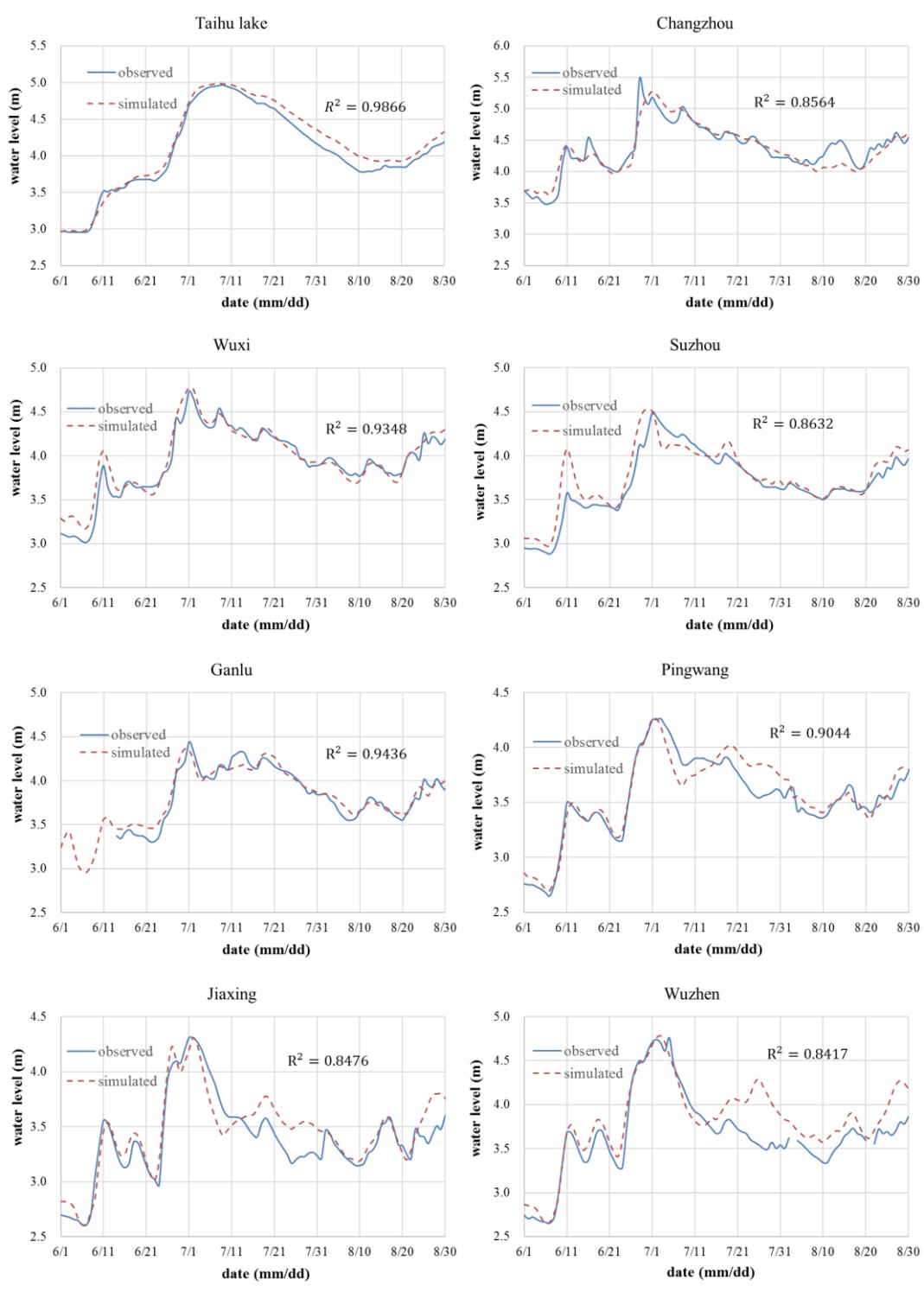

5 **Figure 3: Comparison between the observed and simulated water levels from June to August in the 1999 flood event at eight stations as shown in Figure 1 (adapted from Ou and Wu, 2001)**

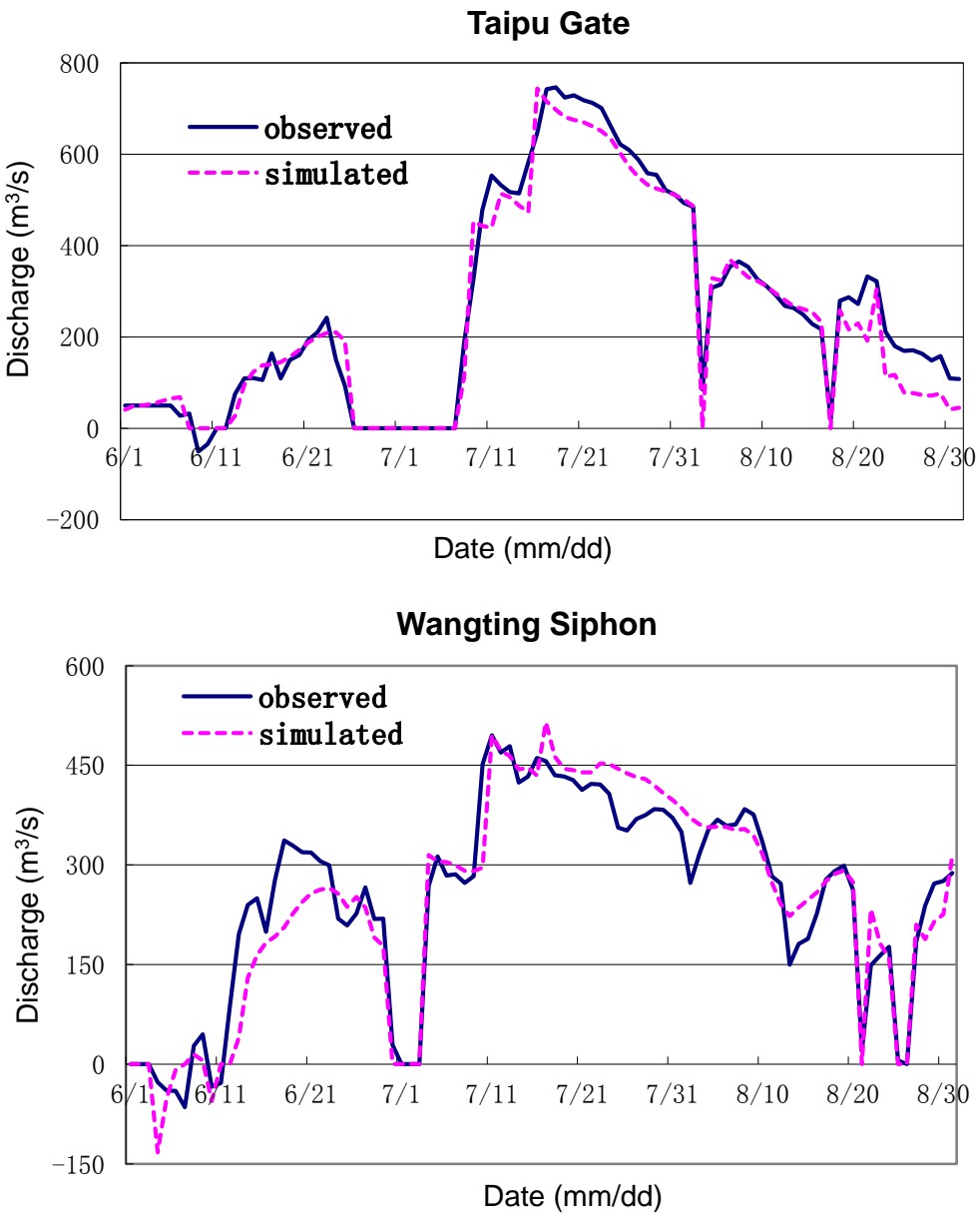

**Figure 4: Comparison between the observed and simulated daily discharges from June to August in the 1999 flood event at the Taipu Gate station and Wangting Siphon station (adapted from Ou and Wu, 2001)**

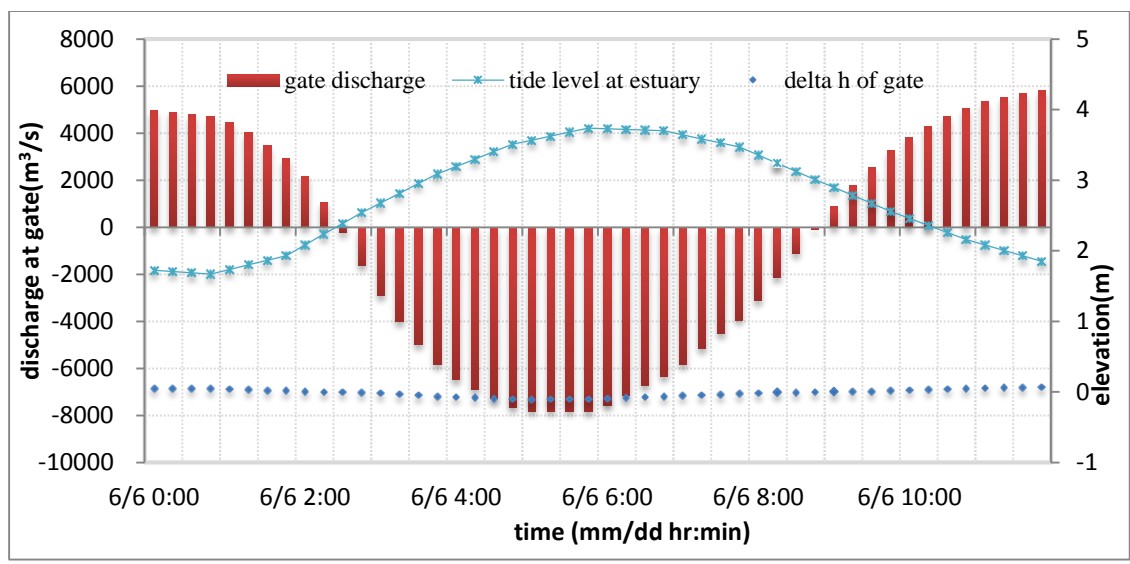

**Figure 5: A test example when the gate keeps open due to the high water lower than tide threshold in this tidal period (t.l.: tide level; w.l. water level; a negative discharge indicates the tidal water intrusions)**

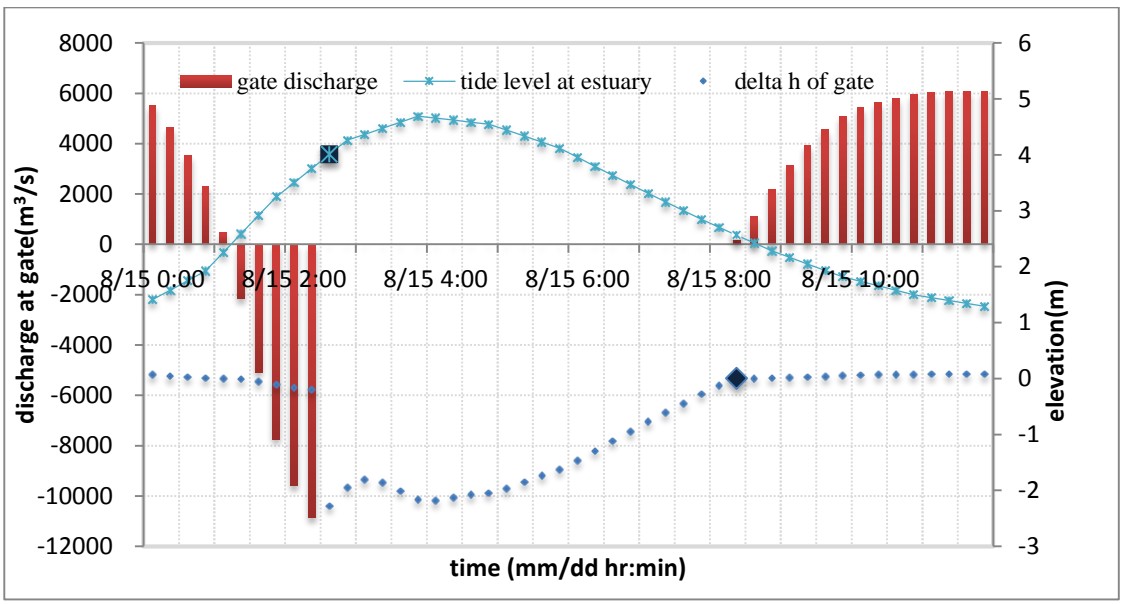

5  **Figure 6: A test example when the gate needs to be closed due to the high water higher than tide threshold in this tidal period (t.l.: tide level; w.l. water level; a negative discharge indicates the tidal water intrusions)**

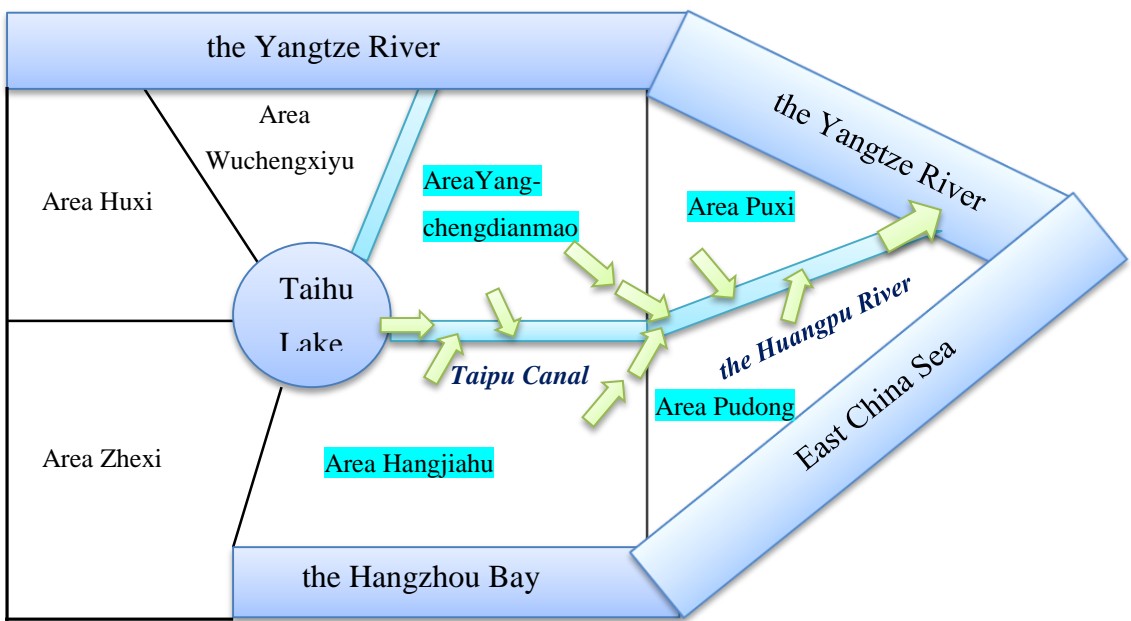

**Figure 7: Conceptual drainage system along Lake Taihu – the Taipu Canal – the Huangpu River – the Yangtze River Estuary**

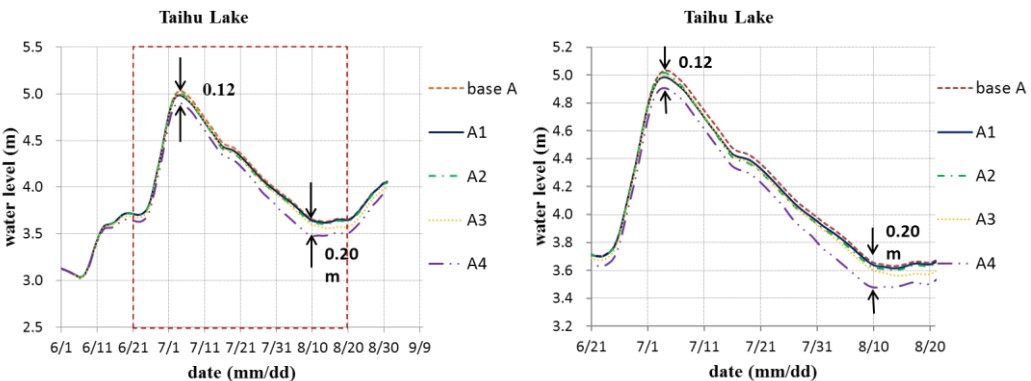

5  **Figure 8: A comparison of the simulated daily lake levels during the period from June to August in 1999 in the Taihu Lake under the five scenarios considered in this study.**

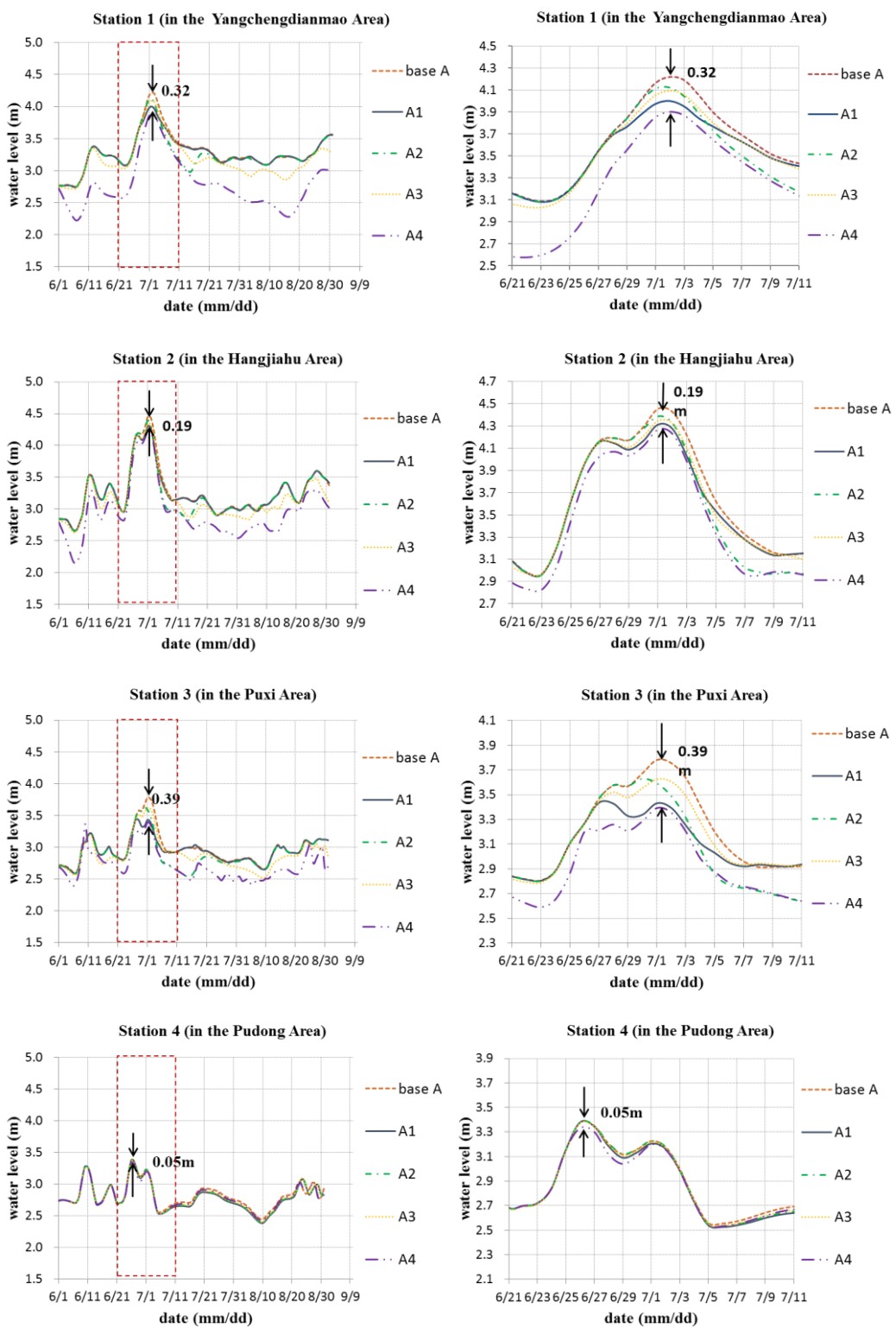

**Figure 9: Comparison of the simulated daily water levels during the period from June to August in 1999 at four stations (as shown in Figure 1) under the five scenarios considered in this study.**

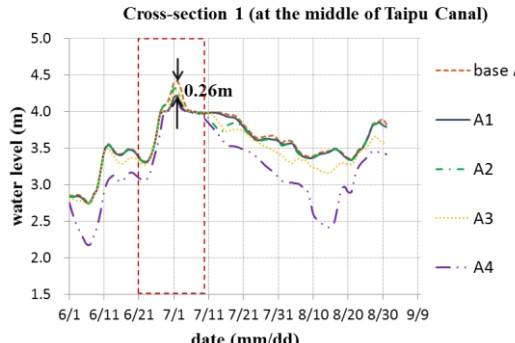

**Cross-section 1 (at the middle of Taipu Canal)**

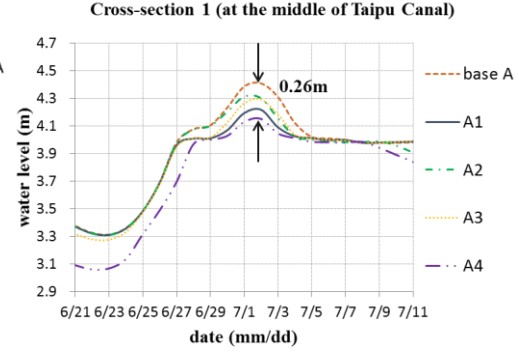

**Cross-section 1 (at the middle of Taipu Canal)**

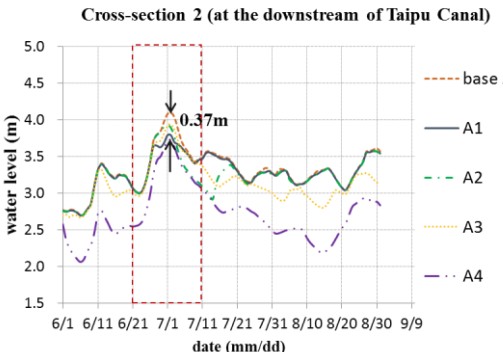

**Cross-section 2 (at the downstream of Taipu Canal)**

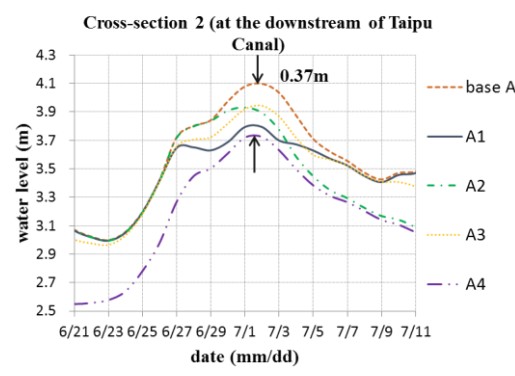

**Cross-section 2 (at the downstream of Taipu Canal)**

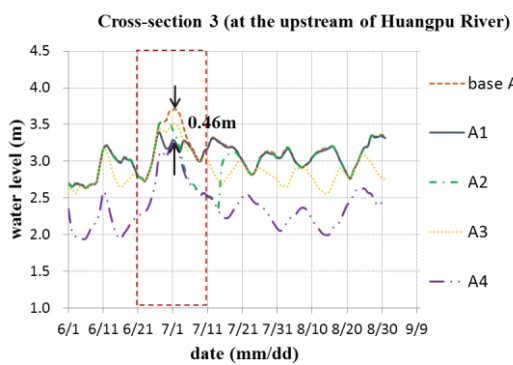

**Cross-section 3 (at the upstream of Huangpu River)**

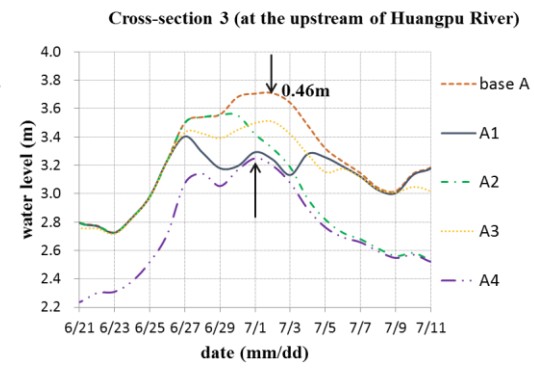

**Cross-section 3 (at the upstream of Huangpu River)**

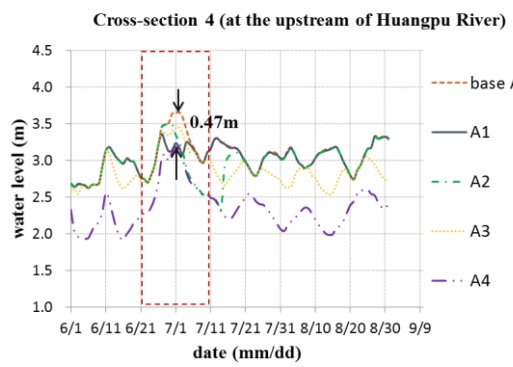

**Cross-section 4 (at the upstream of Huangpu River)**

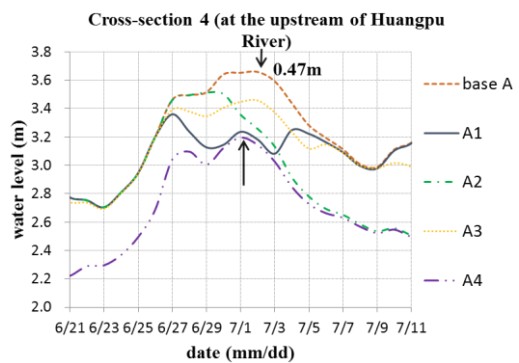

**Cross-section 4 (at the upstream of Huangpu River)**

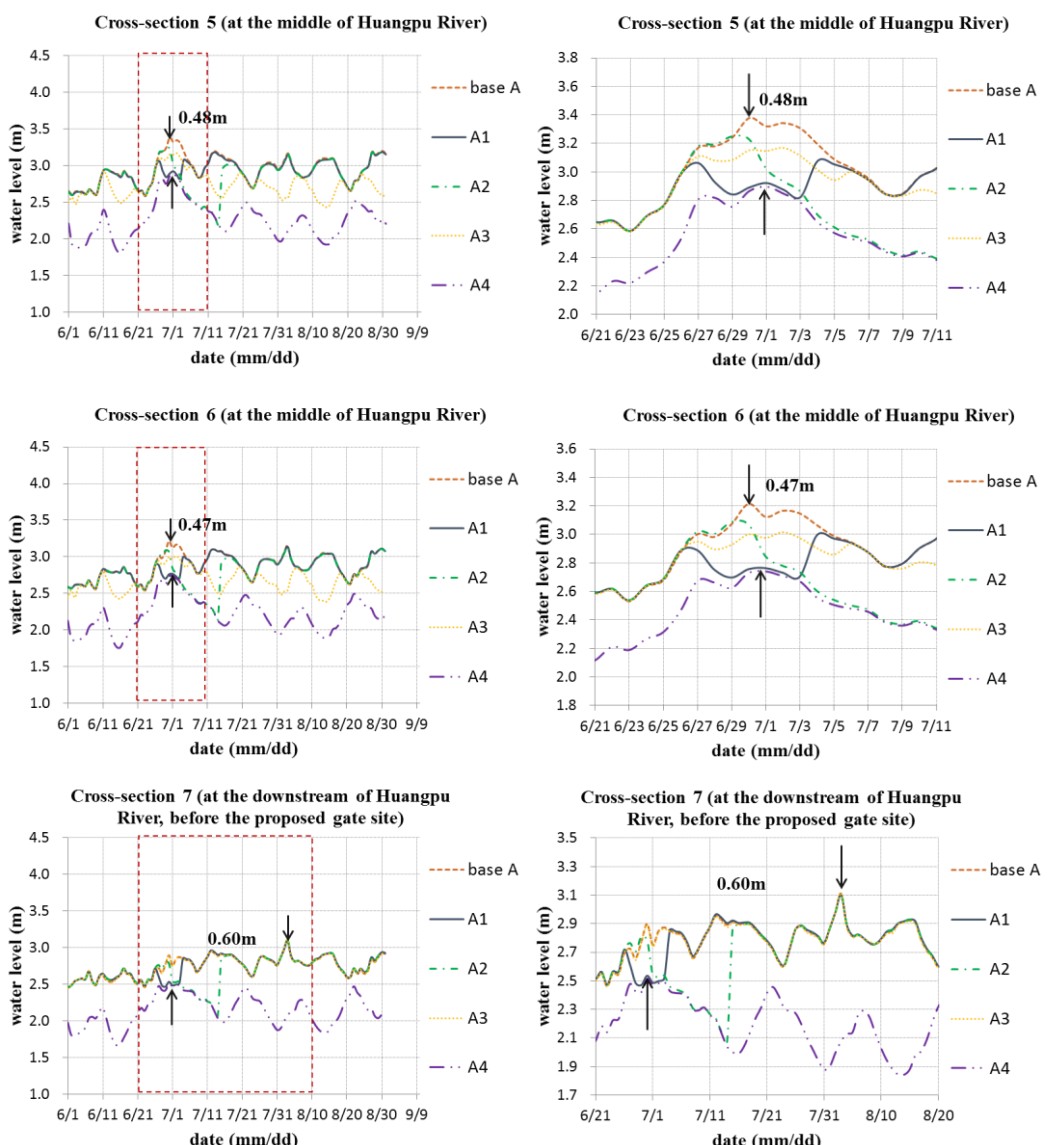

**Figure 10: Comparison of the simulated water levels during the period from June to August in 1999 at the seven cross-section points (as shown in Figure 1) along the Taipu Canal and Huangpu River under five scenarios considered in this study.**

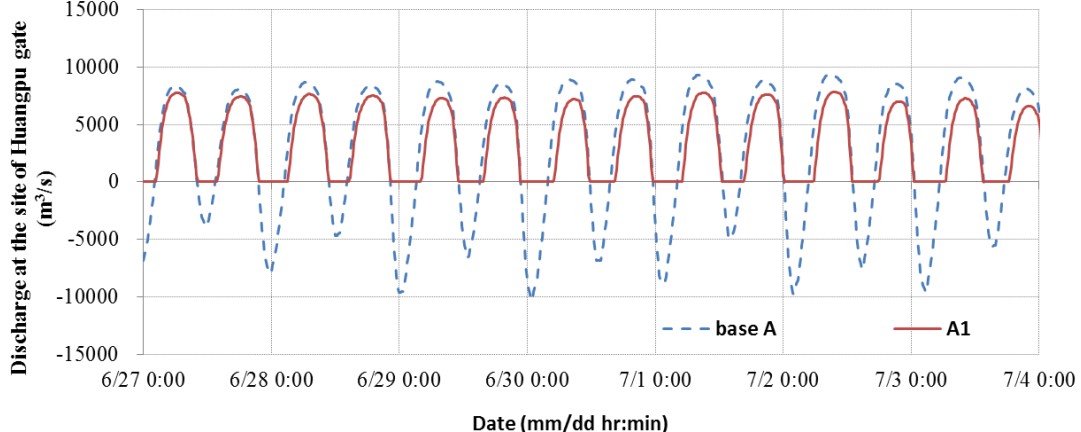

**Figure 11: Comparison of discharges at the site of the proposed gate between the scenarios base A and A1 from June 27 to July 3 in 1999 (The negative discharges means the tidal water intrusion)**

**Table 1: The definitions of five different scenarios considered in this study.**

| Scenario | Definitions |
|---|---|
| base A | without the construction of the proposed gate at the estuary of Huangpu River |
| A1 | with the gate, and operated to prevent tidal intrusion 7 days in advance before the lake level reaches the peaks |
| A2 | with the gate, and operated to prevent water intrusion when the large basin-wide floods occur (defined as the lake level higher than 4.50m) |
| A3 | with the gate, and it will not be closed to prevent tidal water intrusion until the tide rises to a pre-defined threshold (defined as 4.0m in this study) |
| A4 | with the gate, and it will always be closed whenever the tidal water intrudes |

**Table 2: Peak lake water levels and the duration (the number of days) from June to August of 1999 when the lake water levels are higher than a certain control level under five scenarios considered in this study**

| Scenario | peak value. (m) | flood control level 3.5m (days) | high water level 4.0m (days) | design water level 4.65m (1/50) (days) | design water level 4.8m (1/100) (days) |
|---|---|---|---|---|---|
| base A | 5.03 | 81 | 37 | 12 | 8 |
| A1 | 4.99 | 81 | 35 | 11 | 8 |
| A2 | 5.02 | 81 | 34 | 11 | 8 |
| A3 | 5.00 | 81 | 31 | 11 | 8 |
| A4 | 4.91 | 70 | 28 | 10 | 6 |

**Table 3: Peak water levels at the four representative stations under five scenarios considered in this study (unit: m)**

| Scenario | Station 1 | Station 2 | Station 3 | Station 4 |
|---|---|---|---|---|
| base A | 4.22 | 4.46 | 3.78 | 3.38 |
| A1 | 4.00 | 4.31 | 3.43 | 3.38 |
| A2 | 4.12 | 4.39 | 3.63 | 3.38 |
| A3 | 4.09 | 4.35 | 3.62 | 3.37 |
| A4 | 3.90 | 4.27 | 3.39 | 3.33 |

**Table 4: Summary of the inflow volumes of the tributaries in the upstream of the Huangpu River from June to August in 1999 under five scenarios considered in this study (unit: billion m³)**

| Scenario | | base A | A1 | A2 | A3 | A4 |
|---|---|---|---|---|---|---|
| Tributaries in the upstream area of the Huangpu River | outlet of the Taipu Canal | 3.93 | 3.99 | 4.11 | 4.43 | 5.00 |
| | tributaries from the sub-area, northwest of the Huangpu River | 0.59 | 0.62 | 0.64 | 0.77 | 1.05 |
| | tributaries from the sub-area, southwest of the Huangpu River | 1.50 | 1.63 | 1.83 | 2.24 | 3.25 |

**Table 5: Summary of tide intrusion and outflow volume at the site of the proposed gate from June to August in 1999 under five scenarios considered in this study (unit: billion m³)**

| Scenario | Outflow volume at the gate site | | | Times to close the gate | Special explanation about the gate close rules |
|---|---|---|---|---|---|
| | tide intrusion | total outflow volume | net outflow volume | | |
| base A | 17.49 | 24.69 | 7.20 | / | |
| A1 | 15.58 | 23.03 | 7.45 | 14 | from Jun. 27th to Jul. 3rd |
| A2 | 14.14 | 21.94 | 7.80 | 30 | from Jun. 30th to Jul.14th |
| A3 | 10.78 | 19.58 | 8.80 | 74 | until high tide rises up to 4.0m |
| A4 | 0 | 10.95 | 10.95 | 184 | from Jun. 1st to Aug. 31st |