# Peer review of "Numerical Simulations of Potential Contribution of the Proposed Huangpu Gate to Flood Control in the Taihu Lake Basin of China"

_Hydrology and Earth System Sciences, 2016_

## Referee Comment (RC1) · Anonymous Referee #1 · 17 Aug 2016

**Review of** "Model study on potential contributions of the proposed Huangpu Gate to flood control in Taihu Lake basin"

**Overall Recommendation: Major Revision**

**General Comments:**

This manuscript conducted model simulations to investigate the potential contribution/effects of the proposed Huangpu gate for flood control in Taihu Lake basin under various flooding scenarios. Results demonstrate that the proposed gate is more effective to reduce the lake levels compared to the natural channel when the tidal water enters the upper estuary. Based on the scenario analysis, the authors found that the potential contribution of the gate mainly depends on how long the gate operates. In general, the results obtained are well elaborated and the overall conclusions are sound. However, the description of the methodology still needs improvement as the authors did not mention how the model was modified to consider the effects of the proposed gate. Will the simulation results be parameter dependent? What's more, a very important aspect but the authors did not address in the manuscript is the uncertainty analysis. Is it possible to assess the uncertainty for the existing results?

Figures in this paper did not meet the publication quality. It is very difficult to get relative information from figures due to their low resolution. In addition, the writing of this paper is not satisfying and still needs to be improved as there are a lot of grammar errors/typos and inappropriate wording, which make part of this paper difficult to understand. These are enumerated at the end of this review. It is necessary for the authors to ask a native speaker to professionally proofread the manuscript before submission for next round review.

Based on the above considerations, the current form of this paper is not qualified for publication and requires substantial work to improve both the writing and analysis. I recommend to return this manuscript for major revision.

**Specific Comments:**

(1)  Page 1 Line 26: what is the meaning of 327? Is it the page number? Please consider changing the format of the citation. Please also check other places (e.g., Page 3 Line 9).

(2)  Page 2 Line 5: "Balica et al., 2012" cannot be found in the reference list.

(3)  Page 2 Line 25: "Yao & Chen, 1999" cannot be found in the reference list.

(4) Page 3 Line 22: the author mentioned the long-term average. Please specify which period was used to calculate the long-term average.

(5) Page 3 Line 32: flux is defined as the flow per unit area per time, but the unit here for flux is "$m^3$".

(6) Page 4 Line 10: "Duinker and Greig (2007)" cannot be found in the reference list.

(7) Page 4 Line 20: please add the reference for the HOHY model.

(8) In the model description section, the authors mentioned that the original HOHY model was modified to consider the effects of the proposed estuary gate. But there is no information regarding how the modification has been done on the model and what is the main difference between the modified and original model. Please provide more details on this point. In addition, there is no description on which parameters in the model need to be tuned for the calibration.

(9) Page 4 Line 27: please provide more details on the runoff-generation processes for different surface types.

(10) Page 5 Line 3-4: please provide more details on how the water-engineering works are taken into account in the simulation.

(11) The calibration period in this study is from 1984 to 1987, but the verification period is 1995 and 1996. Does the model consider the changes of the underlying surface conditions? e.g., the land use land cover change.

(12) Page 5 Line 17: please specify which period is used to calculate "the peak value of lake level".

(13) Most of the analysis in this study focused on the simulation of lake levels. Is it possible to show how the inundation area is reduced due to the proposed gate?

(14) Figure 1: The quality is very low and it is difficult to figure out the location of stations.

(15) Figure 2 & 3: please (1) increase the resolution of the figures; (2) provide some metrics (e.g., RMSE and $R^2$) to evaluate the model performance; (3) give the unit for the y-axis; (4) in the figure caption, as the observation and simulation have different colors, I prefer to use color instead of "solid"/"dash".

(16) Figure 4 & 5 & 6: please (1) increase the resolution; (2) put a horizontal line indicating the design level in the figure.

(17) Table 2: please specify the date in the caption. Is it 1999?

(18) Table 3: where are these representative stations in Figure 1? What's the unit?

(19) Why chose 7 days in advance for scenario A1? Any particular reasons? Is the number based on some operational rules?

(20) Table 5: (1) how to calculate the times to close the gate? (2) I think the following equation is valid: net outflow = total outflow - tide intrusion. But why the numbers in the table do not meet this equation? Any explanation for this?

**Technical corrections:**

(1) Page 2 Line 8: change "ageing" to "aging".

(2) Page 2 Line 29: change "researches" to "research".

(3) Page 3 Line 4: there should be a space character between the number (36895) and the unit ("km$^2$") similar as Line 12. Please keep this format consistent for other places.

(4) Page 3 Line 8: change "sauce" to "saucer".

(5) Page 3 Line 22: please change "long-term average" to "the long-term average".

(6) Page 3 Line 23: please rephrase "far from the current …". "far from" is difficult to understand.

(7) Page 3 Line 28: change "estuary gate" to "the estuary gate".

(8) Page 4 Line 1-2: please change the format of the citation.

(9) Page 4 Line 8-9: please change "They have since been …" to "Since then, they have been …".

(10) Page 4 Line 9: change "a well-known" to "the well-known".

(11) Page 4 Line 21: change "… gate, and the main Fortran codes of the model is …" to "… gate. The main Fortran codes of the model are …".

(12) Page 4 Line 23: change "stand-alone" to "independently". "stand-alone" is an adjective.

(13) Page 5 Line 1: delete "on".

(14) Page 5 Line 16: change "potential" to "Potential".

(15) Page 6 Line 9: change "potential" to "Potential".

(16) Page 6 Line 15: change "with" to "as".

(17) Page 6 Line 16: change "represent" to "represents".

(18) Page 6 Line 25: change "potential" to "Potential".

(19) Page 6 Line 28: change "was" to "is".

(20) Page 7 Line 8: change "high" to "higher".

(21) Page 7 Line 9: change "are" to "is".

(22) Page 7 Line 13: change "analyses" to "Analyses".

(23) Page 7 Line 14: change "describe" to "describes", change "Rivers" to "River".

(24) Page 7 Line 23: change "describe" to "describes".

(25) Page 7 Line 30: change "impacts" to "impact".

(26) Page 8 Line 11: change "on the different topographies" to "on different topographies".

(27) Page 8 Line 19: change "It is to be noted that …" to "It should be noted that …".

(28) Page 8 Line 20: please rephrase the sentence "…, which make less trouble to the navigation as soon as possible".

(29) Page 8-9: Reference format should be consistent.

(30) Page 14 Table 4: change "summary" to "Summary".

---

## Referee Comment (RC2) · Anonymous Referee #2 · 15 Nov 2016

This manuscript tried to evaluate the potential contributions of the propsed Huangpu Gate to flood control in Taihu Lake basin using a hydraulic model under several flooding scenarios. The results show that the propsed gate is effective mean to evacuate the floodwaters. Maybe it is a useful demonstration of the project. However, the contribution to scientfic progress is not clear, since the method of scenarios analysis is not new and the model is not new.

---

## Short Comment (SC1) · 23 Nov 2016

**Interactive comments on "Model study on potential contributions of the proposed Huangpu Gate to flood control in Taihu Lake basin"**

By Zhang Hanghui et al.

Liu Shuguang et al.

liusgliu@tongji.edu.cn

We appreciate the comments from the reviewer very much, and truly believe these comments can help us to improve the quality of our manuscript. We hope the manuscript after modification would achieve publication status. We provide responses to the main and specific comments and technical corrections in sequential order as follows.

**Specific Comments:**

(1) Page 1 Line 26: what is the meaning of 327? Is it the page number? Please consider changing the format of the citation. Please also check other places (e.g., Page 3 Line 9).

(2) Page 2 Line 5: "Balica et al., 2012" cannot be found in the reference list.

(3) Page 2 Line 25: "Yao & Chen, 1999" cannot be found in the reference list.

(4) Page 3 Line 22: the author mentioned the long-term average. Please specify which period was used to calculate the long-term average.

(5) Page 3 Line 32: flux is defined as the flow per unit area per time, but the unit here for flux is "m3".

(6) Page 4 Line 10: "Duinker and Greig (2007)" cannot be found in the reference list.

(7) Page 4 Line 20: please add the reference for the HOHY model.

(8) In the model description section, the authors mentioned that the original HOHY model was modified to consider the effects of the proposed estuary gate. But there is no information regarding how the modification has been done on the model and what is the main difference between the modified and original model. Please provide more details on this point. In addition, there is no description on which parameters in the model need to be tuned for the calibration.

(9) Page 4 Line 27: please provide more details on the runoff-generation processes for different surface types.

(10) Page 5 Line 3-4: please provide more details on how the water-engineering works are taken into account in the simulation.

(11) The calibration period in this study is from 1984 to 1987, but the verification period is 1995 and 1996. Does the model consider the changes of the underlying surface conditions? e.g., the land use land cover change.

(12) Page 5 Line 17: please specify which period is used to calculate "the peak value of lake level".

(13) Most of the analysis in this study focused on the simulation of lake levels. Is it possible to show how the inundation area is reduced due to the proposed gate?

(14) Figure 1: The quality is very low and it is difficult to figure out the location of stations.

(15) Figure 2 & 3: please (1) increase the resolution of the figures; (2) provide some metrics (e.g., RMSE and R2) to evaluate the model performance; (3) give the unit for the y-axis; (4)in the figure caption, as the observation and simulation have different colors, I prefer to use color instead of "solid"/"dash".

(16) Figure 4 & 5 & 6: please (1) increase the resolution; (2) put a horizontal line indicating the design level in the figure.

(17) Table 2: please specify the date in the caption. Is it 1999?

(18) Table 3: where are these representative stations in Figure 1? What's the unit?

(19) Why chose 7 days in advance for scenario A1? Any particular reasons? Is the number based on some operational rules?

(20) Table 5: (1) how to calculate the times to close the gate? (2) I think the following equation is valid: net outflow = total outflow - tide intrusion. But why the numbers in the table do not meet this equation? Any explanation for this?

**Responses to Specific Comments:**

(1) Page 1 Line 26: what is the meaning of 327? Is it the page number? Please consider changing the format of the citation. Please also check other places (e.g., Page 3 Line 9).

>*Yes, 327 is the page number. We have changed the format of the citation of our manuscript.*

(2) Page 2 Line 5: "Balica et al., 2012" cannot be found in the reference list.

>*"Balica et al., 2012" is in Page 9 Line 18 of our original manuscript. We have exchange the order of the first author's family name and given name. Please find it in Page 10 Line 19 of our new manuscript.*

(3) Page 2 Line 25: "Yao & Chen, 1999" cannot be found in the reference list.

>*"Yao & Chen, 1999" has been changed to "Shao and Yao, 1999"(Page 11 Line 27).*

 (4) Page 3 Line 22: the author mentioned the long-term average. Please specify which period was used to calculate the long-term average.

>*The long-term average used is a nearly sixty-year period 1954- 2010. Please find it in Page 3 Line 22.*

(5) Page 3 Line 32: flux is defined as the flow per unit area per time, but the unit here for flux is "m3".

>*We have already deleted this sentence and reorganized. Please find it in Page 3 Line 32 to Page 4 Line 2.*

(6) Page 4 Line 10: "Duinker and Greig (2007)" cannot be found in the reference list.

>*"Duinker and Greig (2007)" is in Page 9 Line 15 of our original manuscript. We have exchange the order of authors' family name and given name. Please find it in Page 10 Line 29 of our new manuscript.*

(7) Page 4 Line 20: please add the reference for the HOHY model.

>*"Cheng et al. (2006)" is the main reference book which provides a lot of information about the HOHY model listed in the reference of our manuscript (Page 10 Line 25). We have added some necessary and important information of this model, including its development process, schematization and application. Please find it in "3.2.1 Model selection" and Figure 2.*

(8) In the model description section, the authors mentioned that the original HOHY model was modified to consider the effects of the proposed estuary gate. But there is no information regarding how the modification has been done on the model and what is the main difference between the modified and

original model. Please provide more details on this point. In addition, there is no description on which parameters in the model need to be tuned for the calibration.

*We have added a lot of modification details of this model, including the flowchart and test of the extended Fortran program. Please find it in "3.2.2 Model extension and test" and Figure 5-7.*

*The model extension focuses on the flood routing part, related to the algorithms of unsteady open channel flow, and the inputs of control rules of the gates related to the tidal conditions. The main program was improved by adding a function to judge the stage of tide before running the gates (i.e. in flood or ebb tide), which makes the specification of the gate's control rules more flexible. The original program is modified according to the flowchart given in Fig. 5.The modified model is tested by using a simple example, where the tide threshold is assumed to be 4.0m, with the simulation results illustrated in Figure.6 and Figure 7.*

(9) Page 4 Line 27: please provide more details on the runoff-generation processes for different surface types.

*"Cheng et al. (2006)" is the main reference book which provides a lot of information about the HOHY model listed in the reference of our manuscript (Page 10 Line 25), more details can be found in Chapter 2 of this book. In addition, "Jin. (2008), page 49-51" also provides the details on the runoff-generation processes for different surface types. Simple explanations are as follows:*

*a) Water Surface*

*Runoff production of water surface is the rainfall excess, which can be expressed as the difference between the precipitation and evaporation.*

*b) Paddy Field*

*The amount of water which paddies need is changing in different growing periods. Its runoff production changes according to the previous water level.*

*c) Non-irrigated Farmland*

*As a plain area with abundant river networks, its ground water table is comparatively high. A model named "runoff yield under saturated storage" is used for calculation.*

*d) Constructed Ground*

*These grounds are weak at infiltration. Their runoff production can be simplification as the product of the precipitation and coefficient.*

(10) Page 5 Line 3-4: please provide more details on how the water-engineering works are taken into account in the simulation.

*"Cheng et al. (2006)" is the main reference book which provides a lot of information about the HOHY model listed in the reference of our manuscript (Page 10 Line 25), more details can be found in Chapter 4 of this book. In addition,"Jin. (2008), page 53-55" also provides the details on the simulation of water-engineering works. Simple explanations are as follows:*

(11) The calibration period in this study is from 1984 to 1987, but the verification period is 1995 and 1996. Does the model consider the changes of the underlying surface conditions? e.g., the land use land cover change.

*According to the interpretation of the Taihu lake basin in 1985, 1995 and 2000 by Nanjing Institute of Geography and Limnology, Chinese Academy of Sciences, the land use changed little between the end of the 1980s and the beginning of the 1990s.*

[Figure]

the year of 1985            the year of 1995

The year of 2000

(12) Page 5 Line 17: please specify which period is used to calculate "the peak value of lake level".

*The period used to calculate the peak value of lake level from June 1st to August 31$^{st}$, 1999, which was also mentioned in "Page 5 Line 18" of our original manuscript.*

(13) Most of the analysis in this study focused on the simulation of lake levels. Is it possible to show how the inundation area is reduced due to the proposed gate?

*It is a pity we cannot provide the inundation area reduced due to the proposed gate. In generally, the inundation area is calculated by 2-D hydrodynamic model while the HOHY model is a hydrodynamic model for 1-D unsteady open channel flow.*

(14) Figure 1: The quality is very low and it is difficult to figure out the location of stations.

*We have redrawn all figures of our manuscript. Please find the new Figure 1.*

(15) Figure 2 & 3: please (1) increase the resolution of the figures; (2) provide some metrics (e.g., RMSE and R2) to evaluate the model performance; (3) give the unit for the y-axis; (4)in the figure caption, as the observation and simulation have different colors, I prefer to use color instead of "solid"/"dash".

*(1)We have redrawn all figures of our manuscript. Please find them in the new Figure 3-4. (2)We have added RMSE of curves in the new Figure3-4. (3) We have added it. (4)We have updated the figures.*

(16) Figure 4 & 5 & 6: please (1) increase the resolution; (2) put a horizontal line indicating the design level in the figure.

*We have redrawn all figures of our manuscript. Please find them in new Figure 9-11. Besides, we have added the warning levels in the figure, which can properly represent the flood control situation in those areas*

(17) Table 2: please specify the date in the caption. Is it 1999?

*Yes, it is 1999. We have added the date in the caption of Table 2. Please find it in the new Table 2.*

(18) Table 3: where are these representative stations in Figure 1? What's the unit?

*Figure 1 not only gives the location of the Taihu lake basin, but also gives the locations of the four representative stations, which are used to analyze the contributions of the gate to the vulnerable areas. So these locations have no unit.*

(19) Why chose 7 days in advance for scenario A1? Any particular reasons? Is the number based on some operational rules?

*In the Taihu lake basin, a big basin-wide flood means its return year is between 1 in 20 years and 1 in 50 years. More specifically, when the lake level is up to 4.50m or the average rainfall amount of the whole basin in maximum 30 days is up to 450mm. In Scenario A1' means the proposed gate will be operated in the rising stage of the lake levels with a high possibility to create new record of the lake level based on weather forecast. In the simulation of 1999 flood event, there is about one week before lake level reaches its peak value. Therefore, the estuary gate is to be operated 7 days in advance. Please find the comments to this question in "Page 4 Line 19-21" of our new manuscript.*

(20) Table 5: (1) how to calculate the times to close the gate? (2) I think the following equation is valid: net outflow = total outflow - tide intrusion. But why the numbers in the table do not meet this equation? Any explanation for this?

*(1) Simulation results can provide the discharges at any cross-section of rivers, and the times to close the gate in Table 5 equals to the count of discharge change from non-zero to zero; (2) There is a little difference in the equation you mentioned, which was caused by a statistical error. We have corrected it, please find Table 5.*

**Technical corrections:**

(1) Page 2 Line 8: change "ageing" to "aging".

(2) Page 2 Line 29: change "researches" to "research".

(3) Page 3 Line 4: there should be a space character between the number (36895) and the unit ("$km^2$") similar as Line 12. Please keep this format consistent for other places.

(4) Page 3 Line 8: change "sauce" to "saucer".

(5) Page 3 Line 22: please change "long-term average" to "the long-term average".

(6) Page 3 Line 23: please rephrase "far from the current …". "far from" is difficult to understand.

(7) Page 3 Line 28: change "estuary gate" to "the estuary gate".

(8) Page 4 Line 1-2: please change the format of the citation.

(9) Page 4 Line 8-9: please change "They have since been …" to "Since then, they have been …".

(10) Page 4 Line 9: change "a well-known" to "the well-known".

(11) Page 4 Line 21: change "… gate, and the main Fortran codes of the model is …" to "…gate. The main Fortran codes of the model are …".

(12) Page 4 Line 23: change "stand-alone" to "independently". "stand-alone" is an adjective.

(13) Page 5 Line 1: delete "on".

(14) Page 5 Line 16: change "potential" to "Potential".

(15) Page 6 Line 9: change "potential" to "Potential".

(16) Page 6 Line 15: change "with" to "as".

(17) Page 6 Line 16: change "represent" to "represents".

(18) Page 6 Line 25: change "potential" to "Potential".

(19) Page 6 Line 28: change "was" to "is".

(20) Page 7 Line 8: change "high" to "higher".

(21) Page 7 Line 9: change "are" to "is".

(22) Page 7 Line 13: change "analyses" to "Analyses".

(23) Page 7 Line 14: change "describe" to "describes", change "Rivers" to "River".

(24) Page 7 Line 23: change "describe" to "describes".

(25) Page 7 Line 30: change "impacts" to "impact".

(26) Page 8 Line 11: change "on the different topographies" to "on different topographies".

(27) Page 8 Line 19: change "It is to be noted that …" to "It should be noted that …".

(28) Page 8 Line 20: please rephrase the sentence "…, which make less trouble to the navigation as soon as possible".

(29) Page 8-9: Reference format should be consistent.

(30) Page 14 Table 4: change "summary" to "Summary".

*Response to technical corrections:*

*(1) Page 2 Line 8: "ageing" >> Page 2 Line 9: "aging".*

*(2) Page 2 Line 29: "researches" >> Page 2 Line 29: "research".*

*(3) Page 3 Line 4: there should be a space character between the number (36895) and the unit ("$km^2$") similar as Line 12. >> Page 3 Line 4"36895 $km^2$".*

*(4) Page 3 Line 8: "sauce" >> Page 3 Line 8: "saucer".*

*(5) Page 3 Line 22: "long-term average" >> Page 3 Line 22: "the long-term average".*

*(6) Page 3 Line 23: "far from the current …">> Page 3 Line 23: "much higher than the current …".*

*(7) Page 3 Line 28: "estuary gate" >> Page 3 Line 28: "the estuary gate".*

*(8) Page 4 Line 1-2: We have changed the format of the citation. Please find it in Page 4 Line 1-2.*

*(9) Page 4 Line 8-9: "They have since been …" >> Page 4 Line 8-9: "Since then, they have been …".*

*(10) Page 4 Line 9: "a well-known" >> Page 4 Line 9: "the well-known".*

*(11) Page 4 Line 21: "... gate, and the main Fortran codes of the model is ..." >> Page 6 Line 18: "...gate. The main Fortran codes of the model are ...".*

*(12) Page 4 Line 23: "stand-alone" >> Page 5 Line 23: "independently".*

*(13) Page 5 Line 1: delete "on" >> Page 5 Line 32: "on" has been deleted.*

*(14) Page 5 Line 16: "potential" >> Page 7 Line 14: "Potential".*

*(15) Page 6 Line 9: "potential" >> Page 8 Line 6: "Potential".*

*(16) Page 6 Line 15: "with" >> Page 8 Line 12: "as".*

*(17) Page 6 Line 16: "represent" >> Page 8 Line 13: "represents".*

*(18) Page 6 Line 25: "potential" >> Page 8 Line 22: "Potential".*

*(19) Page 6 Line 28: "was" >> Page 8 Line 25: "is".*

*(20) Page 7 Line 8: "high" >> Page 9 Line 5: "higher".*

*(21) Page 7 Line 9: "are" >> Page 9 Line 6: "is".*

*(22) Page 7 Line 13: "analyses" >> Page 9 Line 10: "Analyses".*

*(23) Page 7 Line 14: "describe" >> Page 9 Line 11: "describes"; Page 7 Line 14: "Rivers" >> Page 9 Line 11: "River".*

*(24) Page 7 Line 23: "describe" >> Page 9 Line 20: "describes".*

*(25) Page 7 Line 30: "impacts" >> Page 9 Line 27: "impact".*

*(26) Page 8 Line 11: "on the different topographies" >> Page 10 Line 8: "on different topographies".*

*(27) Page 8 Line 19: "It is to be noted that ..." >> Page 10 Line 16: "It should be noted that ...".*

*(28) Page 8 Line 20: "..., which make less trouble to the navigation as soon as possible" >> "..., making its influences to the shipping much less as soon as possible".*

*(29) Page 8-9: Reference format should be consistent. >>Page 10-11: Reference format has been adjusted.*

*(30) Page 14 Table 4: "summary" >> Page 20 Table 4: "Summary".*

---

## Author Comment (AC1) · 23 Dec 2016

We appreciate your comments very much, and truly believe these comments can help us to improve the quality of our manuscript. Please find our responses and the updated manuscript in attachment. Thank you again.

Please also note the supplement to this comment:
http://www.hydrol-earth-syst-sci-discuss.net/hess-2016-310/hess-2016-310-AC1-supplement.zip

---

## Author Response (AR1)

**Interactive comments on "Model study on potential contributions of the proposed Huangpu Gate to flood control in Taihu Lake basin"**

Zhang Hanghui, Liu Shuguang,

Department of Hydraulic Engineering,

Tongji University, 200092, Shanghai

February 11, 2017

zhang_hanghui@hotmail.com

RE: **hess-2016-310**

Dear Editor:

We appreciate the comments from the reviewer very much, and truly believe these comments can help us to improve the quality of our manuscript. We hope the manuscript after modification would achieve publication status. We provide responses to the main and specific comments and technical corrections in sequential order as follows. Besides, we change the title from "Model study on …" to "Numerical analysis of …". We also make efforts to correct the mistakes and improve the English of the manuscript. All the changes made in the revised manuscript are marked-up in red.

Best regards.

Yours sincerely,

Zhang Hanghui and Liu Shuguang

The following is a point-by-point response to the reviewers' comments.

**Part I Responses to Referee #1**

**Responses to Specific Comments:**

**Q(1)** Page 1 Line 26: what is the meaning of 327? Is it the page number? Please consider changing the format of the citation. Please also check other places (e.g., Page 3 Line 9).

*A: Yes, 327 is the page number. We have changed the format of the citation of our manuscript.*

**Q(2)** Page 2 Line 5: "Balica et al., 2012" cannot be found in the reference list.

*A: "Balica et al., 2012" is in Page 9 Line 18 of our original manuscript. We have exchange the order of the first author's family name and given name. Please find it in Page 10 Line 28 of our new manuscript.*

**Q(3)** Page 2 Line 25: "Yao & Chen, 1999" cannot be found in the reference list.

*A: "Yao & Chen, 1999" has been changed to "Shao and Yao, 1999"(Page 12 Line 1).*

**Q(4)** Page 3 Line 22: the author mentioned the long-term average. Please specify which period was used to calculate the long-term average.

*A: The long-term average used is a nearly sixty-year period 1954- 2010. Please find it in Page 3 Line 27.*

**Q(5)** Page 3 Line 32: flux is defined as the flow per unit area per time, but the unit here for flux is "m3".

*A: We have already deleted this sentence and reorganized. Please find it in Page 3 Line 39 to Page 4 Line 2.*

**Q(6)** Page 4 Line 10: "Duinker and Greig (2007)" cannot be found in the reference list.

*A: "Duinker and Greig (2007)" is in Page 9 Line 15 of our original manuscript. We have exchange the order of authors' family name and given name. Please find it in Page 11 Line 4 of our new manuscript.*

**Q(7)** Page 4 Line 20: please add the reference for the HOHY model.

*A: "Cheng et al. (2006)" is the main reference book which provides a lot of information about the HOHY model listed in the reference of our manuscript (Page 10 Line 35). We have added some necessary and important information of this model, including its development process, schematization and application. Please find it in "3.2 Model development" and Figure 2.*

**Q(8)** In the model description section, the authors mentioned that the original HOHY model was modified to consider the effects of the proposed estuary gate. But there is no information regarding how the modification has been done on the model and what is the main difference between the modified and original model. Please provide more details on this point. In addition, there is no description on which parameters in the model need to be tuned for the calibration.

*A: We have added a lot of modification details of this model, including the flowchart and test of the extended Fortran program. Please find it in "3.2 Model development" and Figure 2, 5-6.*

*The model extension focuses on the flood routing part, related to the algorithms of unsteady open channel flow, and the inputs of control rules of the gates related to the tidal conditions. The main program was improved by adding a function to judge the stage of tide before running the gates (i.e. in flood or ebb tide), which makes the specification of the gate's control rules more flexible. The original program is modified according to the flowchart given in Fig. 2.The modified model is tested by using a*

*simple example, where the tide threshold is assumed to be 4.0m, with the simulation results illustrated in Figure.5 and Figure 6.*

**Q(9)** Page 4 Line 27: please provide more details on the runoff-generation processes for different surface types.

*A:"Cheng et al. (2006)" is the main reference book which provides a lot of information about the HOHY model listed in the reference of our manuscript (Page 10 Line 35), more details can be found in Chapter 2 of this book. In addition,"Jin. (2008), page 49-51" also provides the details on the runoff-generation processes for different surface types. Simple explanations are as follows:*

*a) Water Surface*

*Runoff production of water surface is the rainfall excess, which can be expressed as the difference between the precipitation and evaporation.*

*b) Paddy Field*

*The amount of water which paddies need is changing in different growing periods. Its runoff production changes according to the previous water level.*

*c) Non-irrigated Farmland*

*As a plain area with abundant river networks, its ground water table is comparatively high. A model named "runoff yield under saturated storage" is used for calculation.*

*d) Constructed Ground*

*These grounds are weak at infiltration. Their runoff production can be simplification as the product of the precipitation and coefficient.*

**Q(10)** Page 5 Line 3-4: please provide more details on how the water-engineering works are taken into account in the simulation.

*A: "Cheng et al. (2006)" is the main reference book which provides a lot of information about the HOHY model listed in the reference of our manuscript (Page 10 Line 35), more details can be found in Chapter 4 of this book. In addition,"Jin. (2008), page 53-55" also provides the details on the simulation of water-engineering works.*

*In the Taihu Basin, hydraulic structures are mainly weirs, sluice gates, and pump stations. Different structures have different governing equations. For example, simple explanations are as follows:*

*a) Broad-crested weir:*

*Free flow:* $Q = \mu e B \sqrt{2g(H_0 - h_t)}$

*Submerged flow:* $Q = \varphi_m B(Z_2 - Z_d)\sqrt{2g(Z_1 - Z_2)}$

*b) Broad-crested weir with sluice gate:*

[Figure]

[Figure]

**Schematic Diagram of "Broad-Crested Weir with Sluice Gate"**

*Free flow:* $Q = \varepsilon'\varphi eB\sqrt{2g(H_0 - h_{co})}$

*Submerged flow:* $Q = \mu eB\sqrt{2g(H_0 - h_t)}$

**c) Practical weir with sluice gate:**

[Figure]

**Schematic Diagram of "Practical Weir with Sluice Gate"**

*Free flow:*$Q = \mu_1 eB\sqrt{2gH_0}$

*Submerged flow:*$Q = \mu_1 eB\sqrt{2g(H_0 - h_s)}$

**Q(11)** The calibration period in this study is from 1984 to 1987, but the verification period is 1995 and 1996. Does the model consider the changes of the underlying surface conditions? e.g., the land use land cover change.

*A: According to the interpretation of the Taihu lake basin in 1985, 1995 and 2000 by Nanjing Institute of Geography and Limnology, Chinese Academy of Sciences, the land use changed little between the end of the 1980s and the beginning of the 1990s.*

[Figure]

the year of 1985                                   the year of 1995

The year of 2000

**Q(12)** Page 5 Line 17: please specify which period is used to calculate "the peak value of lake level".

*A: The period used to calculate the peak value of lake level from June 1st to August 31$^{st}$, 1999, which was also mentioned in "Page 5 Line 18" of our original manuscript.*

**Q(13)** Most of the analysis in this study focused on the simulation of lake levels. Is it possible to show how the inundation area is reduced due to the proposed gate?

*A: It is a pity we cannot provide the inundation area reduced due to the proposed gate. In generally, the inundation area is calculated by 2-D hydrodynamic model while the HOHY model is a hydrodynamic model for 1-D unsteady open channel flow.*

**Q(14)** Figure 1: The quality is very low and it is difficult to figure out the location of stations.

*A: We have redrawn all figures of our manuscript. Please find the new Figure 1.*

**Q(15)** Figure 2 & 3: please (1) increase the resolution of the figures; (2) provide some metrics (e.g., RMSE and R2) to evaluate the model performance; (3) give the unit for the y-axis; (4)in the figure caption, as the observation and simulation have different colors, I prefer to use color instead of "solid"/"dash".

*A: (1)We have redrawn all figures of our manuscript. Please find them in the new Figure 3-4. (2)We have added RMSE of curves in the new Figure3. (3) We have added it. (4)We have updated the figures.*

**Q(16)** Figure 4 & 5 & 6: please (1) increase the resolution; (2) put a horizontal line indicating the design level in the figure.

*A: We have redrawn all figures of our manuscript. Please find them in new Figure 8-10. In Taihu Basin, warning levels of the stations are always used to represent the flood control situation in those areas and the design level for the rivers almost cannot be mentioned.*

**Q(17)** Table 2: please specify the date in the caption. Is it 1999?

*A: Yes, it is 1999. We have added the date in the caption of Table 2. Please find it in the new Table 2.*

**Q(18)** Table 3: where are these representative stations in Figure 1? What's the unit?

*A: Figure 1 not only gives the location of the Taihu lake basin, but also gives the locations of the four representative stations, which are used to analyze the contributions of the gate to the vulnerable areas. So these locations have no unit. Please find them in the new Figure 1 (b) and (c).*

**Q(19)** Why chose 7 days in advance for scenario A1? Any particular reasons? Is the number based on some operational rules?

*A: In Taihu basin, a big basin-wide flood means its return year is between 1 in 20 years and 1 in 50 years. More specifically, when the lake level is up to 4.50m or the average rainfall amount of the whole basin in maximum 30 days is up to 450mm. In Scenario A1' means the proposed gate will be operated in the rising stage of the lake levels with a high possibility to create new record of the lake level based on weather forecast. In the simulation of 1999 flood event, there is about one week before lake level reaches its peak value. Therefore, the estuary gate is to be operated 7 days in advance. Please find the comments to this question in "Page 4 Line 23-25" of our new manuscript.*

**Q(20)** Table 5: (1) how to calculate the times to close the gate? (2) I think the following equation is valid: net outflow = total outflow - tide intrusion. But why the numbers in the table do not meet this equation? Any explanation for this?

*A: (1) Simulation results can provide the discharges at any cross-section of rivers, and the times to close the gate in Table 5 means the count of discharge change from non-zero to zero; (2) There is a little difference in the equation you mentioned, which was caused by a statistical error. We have corrected it, please find Table 5.*

*Response to technical corrections:*

*(1) Page 2 Line 8: "ageing" >> Page 2 Line 10: "aging".*
*(2) Page 2 Line 29: "researches" >> Page 2 Line 37: "research".*

*(3) Page 3 Line 4: there should be a space character between the number (36895) and the unit ("$km^2$") similar as Line 12. >> Page 3 Line 5"36895 $km^2$".*

*(4) Page 3 Line 8: "sauce" >> Page 3 Line 9: "saucer".*

*(5) Page 3 Line 22: "long-term average" >> Page 3 Line 26: "the long-term average".*

*(6) Page 3 Line 23: "far from the current ..."  >> Page 3 Line 28: "much higher than the current ...".*

*(7) Page 3 Line 28: "estuary gate" >> Page 3 Line 35: "the estuary gate".*

*(8) Page 4 Line 1-2: We have changed the format of the citation. Please find it in Page 3 Line 39- Page 4 Line2.*

*(9) Page 4 Line 8-9: "They have since been ..." >> Page 4 Line 10: "Since then, they have been ...".*

*(10) Page 4 Line 9: "a well-known" >> Page 4 Line 10: "the well-known".*

*(11) Page 4 Line 21: "... gate, and the main Fortran codes of the model is ..." >> Page 6 Line 19: "...gate. The main Fortran codes of the model are ...".*

*(12) Page 4 Line 23: "stand-alone" >> Page 5 Line 26: "independently".*

*(13) Page 5 Line 1: delete "on" >> Page 5 Line 37: "on" has been deleted.*

*(14) Page 5 Line 16: "potential" >> Page 7 Line 14: "Potential".*

*(15) Page 6 Line 9: "potential" >> Page 8 Line 6: "Potential".*

*(16) Page 6 Line 15: "with" >> Page 8 Line 13: "as".*

*(17) Page 6 Line 16: "represent" >> Page 8 Line 14: "represents".*

*(18) Page 6 Line 25: "potential" >> Page 8 Line 26: "Potential".*

*(19) Page 6 Line 28: "was" >> Page 8 Line 30: "is".*

*(20) Page 7 Line 8: "high" >> Page 9 Line 7: "higher".*

*(21) Page 7 Line 9: "are" >> Page 9 Line 8: "is".*

*(22) Page 7 Line 13: "analyses" >> Page 9 Line 13: "Analyses".*

*(23) Page 7 Line 14: "describe" >> Page 9 Line 14: "describes"; Page 7 Line 13: "Rivers" >> Page 9 Line 13: "River".*

*(24) Page 7 Line 23: "describe" >> Page 9 Line 24: "describes".*

*(25) Page 7 Line 30: "impacts" >> Page 9 Line 33: "influence".*

*(26) Page 8 Line 11: "on the different topographies" >> Page 10 Line 14: "on different topographies".*

*(27) Page 8 Line 19: "It is to be noted that ..." >> Page 10 Line 25: "much attention should be paid to ...".*

*(28) Page 8 Line 20: "..., which make less trouble to the navigation as soon as possible" >> Page 10 Line 24-25: "When the operation rules of the proposed gate is formulated, much attention should be paid to the navigation in the river to mitigate the influence on the shipping as less as possible".*

*(29) Page 8-9: Reference format should be consistent. >>Page 10-12: Reference format has been adjusted.*

*(30) Page 14 Table 4: "summary" >> Page 22 Table 4: "Summary".*

**Part II Responses to Referee #2**

**Main Comments:**

This manuscript tried to evaluate the potential contributions of the proposed Huangpu Gate to flood control in Taihu Lake basin using a hydraulic model under several flooding scenarios. The results show that the proposed gate is effective mean to evacuate the floodwaters. Maybe it is a useful demonstration of the project. However, the contribution to scientific progress is not clear, since the method of scenarios analysis is not new and the model is not new.

**Responses to Main Comments:**

Like the Thames barrier in London, the proposed Huangpu gate also has the function of preventing tide intrusion. Besides, the Huangpu gate also can help the lake and the upstream areas to drain flood water when basin-level floods occur. This paper provides quantitative analyses of the potential benefits of the proposed Huangpu gate when the basin suffers monsoon-induced floods. These conclusions are very important for the basin authority's management.

Although the model used in the paper is not completely new, we modified the model codes to accommodate additional capabilities for the more complex simulation. The Schematization of the modified model is shown in the figure with table gridlines as below. The model extension focuses on the flood routing part, related to the algorithms of unsteady open channel flow, and the inputs of control rules of the gates related to the tidal conditions. The main program was improved by adding a function to judge the stage of tide before running the gates (i.e. in flood or ebb tide), which makes the specification of the gate's control rules more flexible.

[Figure]

The Taihu Lake Basin is typified by a dense water web and a flat saucer-like landform, forming a complex hydro-system that includes interlaced rivers, dense water nets and dotted depression lakes of different sizes. Especially, there are lots of hydraulic structures, such as weirs, sluice gates, and pump stations. Different hydraulic structures have different optional rules. The HOHY model is one of the outcomes of a three-year water quality study in the Taihu lake basin supported by the World Bank loan, which were jointly undertaken by Hohai University and Delft Hydraulics, the Netherlands. The HOHY model can simulate the cycle of flood waters in the basin well. Not only can it simulate complex hydro-systems with numerous interlaced rivers and lakes, and complicated relationships between river

nets, hilly areas and tidal boundaries, it also can simulate complex operational rules of control structures, such as sluices, pumps and siphons. This modified model is based on the features of the Huangpu proposed gate and its multi-functions. It is **this modified model** that can precisely simulate the complex operation rules of the proposed gate, where its operational rules will be applied for the flood tide and ebb tide respectively. As far as I know, very few models can accurately simulate such complex rules of hydraulic structures.

---

## Referee Report (RR1)

**Review of** "Numerical analysis of potential contributions of the proposed Huangpu Gate to flood control in Taihu Lake basin"

**Overall Recommendation: Major Revision**

**General Comments:**

This is a second version of the manuscript that was previously reviewed by this reviewer. The authors made some changes to address my "specific comments". But they did not provide any response to my "general comments". For example, whether the simulation results are parameter dependent and whether it is possible to assess the uncertainty for the existing results.

In addition, I did not see much improvement of the writing compared to the last submission. There are still a lot of grammar errors/typos and inappropriate wording (enumerated at the end of this review). I STRONGLY encourage the authors to ask a native speaker to professionally proofread the manuscript before submission for next round review.

Based on the above considerations, I recommend the manuscript to be returned to the authors for major revision again to respond my general comments and to further improve the language. Detailed comments follow below:

**Specific Comments:**

(1)   Page 6 Line 6: This is not a 4-year period.

(2)   Page 6 Line 8: The authors mentioned that the model was tested for the 1954, 1991 and 1999 flood events, but why only show 1999 in this study?

(3)   Page 10 Line 4: Please be clearer of "The potential contribution".

(4)   Figure 1(b): Please show station names in the map.

(5)   Figure 3: The x-label should include the year. Please also check other figures.

**Technical corrections (Not an Exhaustive List):**

(1)   Page 1 Line 13-14: Change "is dependent on" to "depends on".

(2)   Page 1 Line 15: Change "is a quite … connects" to "plays an important role to connect".

(3)   Page 1 Line 20: Change "The results" to "Results", change "an effective mean" to "effective".

(4)   Page 1 Line 27: Change "flows to" to "flows into".

(5)   Page 1 Line 34: Change "To fight … of flood" to "Traditional methods for flood".

(6)   Page 2 Line 1: Change "operating" to "operated".

(7)   Page 2 Line 6: Please rephrase "empties into".

(8)   Page 2 Line 34-38: Please rephrase "However, …, 2002)."

(9)   Page 3 Line 7: What does "amounts to" mean? Please rephrase and check other places.

(10) Page 3 Line 11: Please rephrase "typified".

(11) Page 3 Line 21: Change "suffer" to "suffering".

(12) Page 3 Line 24: Change "occur" to "occurring".

(13) Page 5 Line 16: Delete "was".

(14) Page 5 Line 24: There are two "in".

(15) Page 10 Line 23: Change "implementations" to "implementation".

---

## Referee Report (RR2)

**Review of the paper "Numerical analysis of potential contributions of the proposed Huangpu gate to flood control in Taihu Lake basin" by Zhang et al.**

In this paper, the authors investigated the potential impacts of constructing a sluice gate (or an estuary gate) on the flood control in terms of evacuating flood discharge and reducing peak water levels, which is indeed important from both scientific and engineering points of view. The possible impacts of different operating mode of the proposed sluice gate on the hydrological conditions in the upstream part of the gate were analyzed based on the numerical simulation for different scenarios. However, the authors focused on the application of HOHY model rather than the analyses of underlying mechanism of different operating modes for flood control. In addition, the authors did not investigate the potential siltation both upstream and downstream of the sluice gate, which is extremely important for real yet practical use of constructing a sluice gate for flood control purpose in an estuary. Based on the major concerns below, I would suggest to reject the paper.

**Major concerns:**

1.  It appears that the authors focused on the application of HOHY model to investigate the potential impacts of different operating modes of the gate on the hydrodynamics in the upper region of the studied area. I would suggest the authors to concentrate on the analysis of the underlying mechanism or link between different operating modes and parameters relating to flood control.

2.  The paper did not mention the potential siltation due to the construction of an estuary gate at all. There are numerous studies on the impact of estuary gate or tidal barriers on hydrodynamics and sedimentation in riverine system and estuaries (e.g., Schmidt et al., 2005; Carroll et al., 2008; Ji et al., 2011; Ji et al., 2016; Zhu et al., 2017). It was shown that the sediments from upstream part could be trapped upstream of an estuary gate and cause problems such as increased bed elevation and reduced water storage volume. Hence sedimentation reduction measures, such as sediment flushing, manual dredging, channel contraction or a combination of flushing and channel contraction, are the major concerns of building such a construction. Hence it is not reliable to have such a conclusion of building an estuary gate on the basis of only hydrodynamics simulations.

3.  It is noted that most of the references are written in Chinese, which is not suitable for publishing a paper in an international Journal, such as HESS.

References:

Carroll, B., Li, M., Pan, S., Wolf, J., Burrows, R., 2008. Morphodynamic Impacts of a Tidal Barrage in the Mersey Estuary. World Scientific Publishing Company, pp. 2743–2755.

Ji, U., Julien, P.Y., Park, S.K., 2011. Sediment Flushing at the Nakdong River Estuary Barrage. J Hydraul Eng-Asce, 137(11): 1522-1535.

Ji, U., Jang, E.K., Kim, G., 2016. Numerical modeling of sedimentation control scenarios in the approach channel of the Nakdong River Estuary Barrage, South Korea. Int J Sediment Res, 31(3): 257-263.

Ji, Y.X., Yang, F., Zhang, H.Y., Lu, Y.J., 2013. A siltation simulation and desiltation measurement study downstream of the Suzhou Creek Sluice, China. China Ocean Eng, 27(6): 781-793.

Schmidt, A., Brudy-Zippelius, T., Kopmann, R., Imiela, M., 2005. Investigations to reduce sedimentation upstream of a barrage on the river Rhine. Wit Trans Ecol Envir, 80: 145-154.

Zhu, Q. et al., 2017. Modeling morphological change in anthropogenically controlled estuaries. Anthropocene, 17: 70-83.

---

## Author Response (AR2)

**Interactive comments on "Numerical analysis of potential contributions of the proposed Huangpu Gate to flood control in Taihu Lake basin"**

Zhang Hanghui, Liu Shuguang,

Department of Hydraulic Engineering,

Tongji University, 200092, Shanghai

May 14, 2017

zhang_hanghui@hotmail.com

RE: **hess-2016-310**

Dear Editor:

We appreciate the comments from the reviewer very much, and truly believe these comments can help us to improve the quality of our manuscript. We hope the manuscript after modification would achieve publication status. We provide responses to the general and specific comments and technical corrections in sequential order as follows. We also make efforts to correct the mistakes and improve the English of the manuscript.

Best regards.

Yours sincerely,

Zhang Hanghui and Liu Shuguang

The following is a point-by-point response to the reviewers' comments.

**Part I Responses to Referee #1**

**General Comments:**

This is a second version of the manuscript that was previously reviewed by this reviewer. The authors made some changes to address my "specific comments". But they did not provide any response to my "general comments". For example, whether the simulation results are parameter dependent and whether it is possible to assess the uncertainty for the existing results.

In addition, I did not see much improvement of the writing compared to the last submission. There are still a lot of grammar errors/typos and inappropriate wording (enumerated at the end of this review). I STRONGLY encourage the authors to ask a native speaker to professionally proofread the manuscript before submission for next round review.

**Responses to General Comments:**

*A: First of all, we feel very sorry we missed one piece of your comments. Regarding your comment whether the simulation results are parameter dependent and whether it is possible to assess the uncertainty for the existing results,our answer is Yes. Of course, the simulation results are related to the parameters,In this paper, we focus on the potential impacts of constructing an estuary gate on the flood control in terms of evacuating flood discharge and reducing peak water levels, which is indeed important from both scientific and engineering points of view. So we did not analyse the sensitivity of parameters.*

*Regarding another important suggestion to professionally proofread the manuscript, we also have improved the English of the manuscript.*

**Responses to Specific Comments:**

**Q(1)** Page 6 Line 6: This is not a 4-year period.

*A: Yes, this is not a 4-year period, and data for the model are two consecutive years. We have corrected this mistake.*

**Q(2)** Page 6 Line 8: The authors mentioned that the model was tested for the 1954, 1991 and 1999 flood events, but why only show 1999 in this study?

*A: As a fact, the model was tested for the1954, 1991 and 1999 flood event. However, we only have the data for 1999 flood event. So the year 1954 and 1991 in Page 6 Line 5 of the new manuscript were deleted.*

**Q(3)** Page 10 Line 4: Please be clearer of "The potential contribution".

*A: We have checked and rephrased the conclusion about the potential contribution. Please find it in the "Conclusions" of our new manuscript.*

**Q(4)** Figure 1(b): Please show station names in the map.

*A: Please find them in the new Figure 1 (a).*

**Q(5)** Figure 3: The x-label should include the year. Please also check other figures.

*A: We have checked the figures and added the information to the figure caption. Please find them in the new Figure 3-4 and 11.*

**Response to technical corrections:**

*(1) Page 1 Line 13-14: Change "is dependent on" to "depends on" >> Page 1 Line 14: "depends on".*

*(2) Page 1 Line 15: Change "is a quite ... connects" to "plays an important role to connect" >>. Page 1 Line 15: "...is an important river in the basin. It connects the Taihu Lake upstream and the Yangtze Estuary downstream,.."*

*(3) Page 1 Line 20: Change "The results" to "Results", change "an effective mean" to "effective" >>. Page 1 Line 21: "Results"; Page 1 Line 21: "effective"*

*(4) Page 1 Line 27: Change "flows to" to "flows into" >>. The sentence has already been deleted.*

*(5) Page 1 Line 34: Change "To fight ... of flood" to "Traditional methods for flood" >>. The sentence has already been deleted.*

*(6) Page 2 Line 1: Change "operating" to "operated" >>. Page 2 Line 29: "operated".*

*(7) Page 2 Line 6: Please rephrase "empties into" >>. The sentence has already been deleted.*

*(8) Page 2 Line 34-38: Please rephrase "However, ..., 2002)." >> Page 3 Line 1-4: This sentence has already been rephrased.*

*(9) Page 3 Line 7: What does "amounts to" mean? Please rephrase and check other places >>. Page 3 Line 2: " is up to...", and the sentence has been rephrased.*

*(10) Page 3 Line 11: Please rephrase "typified" >>. Page 3 Line 16: "is characteristic of...".*

*(11) Page 3 Line 21: Change "suffer" to "suffering" >>. Page 3 Line 26: "suffering".*

*(12) Page 3 Line 24: Change "occur" to "occurring" >>. Page 3 Line 29: "occurring".*

*(13) Page 5 Line 16: Delete "was" >>. Page 5 Line 13: "was tested".*

*(14) Page 5 Line 24: There are two "in" >>: Page 5 Line 20: one "in" has been deleted.*

*(15) Page 10 Line 23: Change "implementations" to "implementation" >>. Page 10 Line 20: "implementation".*

**Part II Responses to Referee #2**

**Main Comments:**

In this paper, the authors investigated the potential impacts of constructing a sluice gate (or an estuary gate) on the flood control in terms of evacuating flood discharge and reducing peak water levels, which is indeed important from both scientific and engineering points of view. The possible impacts of different operating mode of the proposed sluice gate on the hydrological conditions in the upstream part of the gate were analysed based on the numerical simulation for different scenarios. However, the authors focused on the application of HOHY model rather than the analyses of underlying mechanism of different operating modes for flood control. In addition, the authors did not investigate the potential siltation both upstream and downstream of the sluice gate, which is extremely important for real yet practical use of constructing a sluice gate for flood control purpose in an estuary.

*A: First of all, this paper not only investigated the application of the HOHY model, but also expanded the model. Second, the Taihu Lake is upstream of the Huangpu River, and the inflow from the upstream area is the main part for the Huangpu River rather than the tidal water from the Yangtze River downstream. From Table 9-1 (Yan, 1992), the sediment concentration upstream is of 0.049 kg/m³, and that is of 0.213 kg/m³ downstream. Both of them are relatively few. For this reason, the problem caused by sediment and siltation is not serious in the Huangpu River estuary. Also because of this, our research focus on the potential contributions for evacuating flood discharge and reducing flood risk after the construction of the estuary gate. As is known to all, investigating the gate's potential contribution is of great engineering significance for the flood control of Shanghai metropolis and the whole basin. Clearly, it is very useful for flood management of the local authority and basin authority.*

Huangpu River
[Figure]

Table 9-1: the classification of the estuaries in China (from Yan, Yan, 1992)
Yan Kai: China coastal engineering, Hohai University Press, 1992 (in Chinese)

**Q(1)** It appears that the authors focused on the application of HOHY model to investigate the potential impacts of different operating modes of the gate on the hydrodynamics in the upper region of the studied arear. I would suggest the authors to concentrate on the analysis of the underlying mechanism or link between different operating modes and parameters relating to flood control.

*A: We agree that analysis of the underlying mechanism or link between different operating modes is a very important subject. However, from our point of view, it is also important to study the potential evacuation benefits of the proposed Huangpu gate. The reasons are as follows: First, the Huangpu River is an important evacuation waterway in Shanghai metropolis; with rising sea level and increasing human activities, the flood problem of the Huangpu River estuary has become increasingly complex, such as the awkward situation of upstream floods, high tide and local rainstorms occurring simultaneously. Therefore, investigating the gate's potential contribution is of great engineering significance for the flood control of Shanghai metropolis and the Taihu Lake basin. Second, although the Thames Barrier in England and the Delta Storm Surge Barriers in Netherlands have proved to be effective controls at the estuary, they can only serve as a qualitative reference. This paper made a quantitative study with the model, which is not only useful for the estuary management but also useful for flood management. Third, the Huangpu River estuary is multi-functional; it not only can block the high tide of the river, but also can accelerate evacuating upstream flood and local waterlogging. This paper simulated the potential functions of the estuary gate through various scenarios and also expanded the HOHY model; therefore, it is of academic significance.*

**Q(2)** The paper did not mention the potential siltation due to the construction of an estuary gate at all. There are numerous studies on the impact of estuary gate or tidal barriers on hydrodynamics and sedimentation in riverine system and estuaries (e.g. Schmidt et al., 2005; Carroll et al., 2008; Ji et al.. 2011; Ji et al., 2016; Zhu. et al., 2017). It was shown that the sediments from upstream part could be trapped upstream of an estuary gate and cause problems such as increased bed elevation and reduced water storage volume. Hence sedimentation reduction measures, such as sediment flushing, manual dredging, channel contraction or a combination of flushing and channel contraction, are the major concerns of building such a construction. Hence it is not reliable to have such a conclusion of building an estuary gate on the basis of only hydrodynamics simulations.

*A: Sediment and siltation is a common problem in estuarine gates and is a major issue to consider before building a gate. The Huangpu River estuary construction is a very complex issue, involving problems such as sediment, siltation, ecology, shipping and so on. As we said in response to main comments, sediment and siltation is not very serious in this area and will not be included in this paper. We focus on the most important point after the construction of the gate, which is to investigate the potential contributions of constructing such an estuary gate to flood control and who will benefit from the proposed gate. More important, this paper not only makes clear the contributions of the gate to the estuary area, but also its contributions to the Taihu Lake Basin, which is also very important to flood management by Taihu Lake Basin Authority.*

**Q(3)** It is noted that most of the references are written in Chinese, which is not suitable for publishing a paper in an international Journal, such as HESS..

*A: Huangpu River is a very important river in Shanghai, China. Estuary gate construction is still in the preliminary demonstration-of-benefit stage; most scholars studying this subject are Chinese and most literatures are also in Chinese. This amendment, we once again added more English literatures as follows.*

*Wang Liang, CaiYongli, Chen Hongquan, Dag Daler, Zhao Jingmin, and Yang Juan: Flood disaster in Taihu Basin, China: casual chain and policyoption analyses, Environ Earth Sci., DOI 10.1007/s12665-010-0786-x, 1119-1124, 2011*

*Zhou zhengzheng, Liu Shuguang, ZhongGuihui, and Cai Yi:Flood Disaster and Flood Control Measurements in Shanghai. Natural Hazards, ISSN 1527-6988, 2016*

*Hu Qingfang and Wang Yintang: Impact assessment of climate change and human activities on annual highest water level of Taihu Lake. Water Science and Engineering, DOI: 10.3882/j.issn.1674-2370, 2009*

---

## Author Response (AR4)

**Interactive comments on "Numerical simulations of potential contribution of the proposed Huangpu Gate to flood control in Taihu Lake basin of China"**

Zhang Hanghui, Liu Shuguang,

 5 Department of Hydraulic Engineering, Tongji University, 200092, Shanghai August 20, 2017 zhang\_hanghui@hotmail.com RE: hess-2016-310

10

Dear Editor:

We appreciate much the comments from both reviewers. We all agree that these comments indeed help us improve the quality of our manuscript, and hence all comments from Reviewers have been considered

- 15 and incorporated into our newly revised manuscript. In this round of revisions, our manuscript has been through very serious, word-by-word revisions from head to toe. We believe our current manuscript is in a much better shape close to the publication level than all our previous versions. We invite the Editor and both Reviewers have a read on our newly revised version of manuscript. Meanwhile, we have provided point-to-point detailed responses to the Reviewers' comments below. We also made efforts to
- 20 correct the mistakes and improve the English of the manuscript. Finally, we have made some minor changes on the title of this manuscript.

Yours sincerely,

25 Zhang Hanghui and Liu Shuguang On behalf of all co-Authors

**Part I Responses to Referee #1**

**Main Comments:**

Authors refer to two types of floods in the Taihu Lake basin (i.e. monsoon-induced and typhoon-induced floods), and the basin is impaced by both marine (e.g. tides, storm surges) and river flooding &

- 5 waterloddging. The methodology section leaves the reader with an expectation of applications of the HOHY model to investigate the potential impacts of different operating modes of the gate to the flood control of the basin for typical flood events mentioned above, such as the 1999 flood, the 1991 flood. However, this expection goes unfulfilled in the result analysis and discussion sections, and the reader is left wondering why scenario analysis was made only for the 1999 flood event.
- 10

A: Thanks for your comments. The Huangpu River is the main shipping and drainage route to the port city Shanghai in China. The main role of the proposed gate is to block tide intrusion. Considering the shipping function, it is impossible for this gate to operate like other gates along the Yangtze River, which are operated more frequently for other purposes. Moreover, the dike also can protect Shanghai

- 15 when normal floods occur. In this study, the proposed gate will be operated only if the flooding conditions in the Taihu Lake basin are very severe. The 1999 flood is the largest flood in history for the study basin, and the total rainfall during the 43-days monsoon period reached 670 mm, three times more than the long-term (1954-2010) average during the same period. The return period of the 1999 flood event was estimated to be 200 years. From the viewpoint of engineering design as is the
- 20 main purpose of this study, only the most extreme situation such as the 1999 flooding condition is of our main concern. That is the main reason we choose only the 1999 flood event for our model simulation.

**Responses to Specific Comments:**

25

30

Q(1) The introduction should point out the issues you want to study, review the research advances in international hydrology society, and present the main research content of the paper. In this regards, it is suggested for authors to review the techniques and approaches adopted for impact study on flood control by operating estuary barries in Japan, England, Netherland and other countries, and evaluate the innovation or new findings in this applied research. In other words, it should indicate why the reader

should be interested in reading the paper?

A: Thanks for your comments. Yes, we have reorganized the chapter, which can be found on Page 2-3 in the newly revised manuscript.

35 **Q(2)** Detailed about the study area of the Taihu Lake should moved from Introduction to Study Area section.

A: Thanks for your comments. Yes, we have modified this chapter, which can be found on Page 3-5 in the newly revised manuscript.

Q(3) More information about the typical floods in the Taihu Lake basin (e.g. the 1999 flood, the 1991 flood) should be given in Study Area section, including the causes of the flooding (e.g. Rainstorm, astronomical tide, storm surge).

A: Thanks for your comments. Yes, we have added more information about the causes of the 5 flooding, which can be found in the sections of Introduction and the Study Area in the newly revised manuscript.

Q(4) General information about sedimentation in Huangpu river and estuaries is needed to supplement in Study area section.

10

A: Thanks for your comments. Yes, we have added more information about sediment, which can be found on the Page 5 Line 14-17 in the newly revised manuscript.

Q(5) In 3.1 Scenarios description (Page 4 Line 20-35), authors describes five scenarios used in the study.
 The reader wonder about the criteria or principles of proposing the five scenarios. What are the main
 factors to be considered in the gate operation scenarios design?

A: Thanks for your comments. The scenario "base A" means the estuary gate would not be constructed at the outlet of the Huangpu River, which is used as a basis for comparison. Scenario "A1" and "A2" are designed to quantitatively analyze its benefits of draining the floodwater from the Taihu lake and the upstream areas. Scenario "A3" is to analyze quantitatively its role to block tide intrusion.

20 The last scenario is just a case to calculate the potential maximum benefits to flood control of the basin.

**Q**(6) In 3.2 Model description (Page 5 Line 12-35), authors should also say if this is the first presentation on the HOHY model in an international Journal. A brief introduction on the HOHY model's structure and methodology, and further development of the model by authors in this study are also recommended

to be added in the paper.

25

30

A: Thanks for your comments. As for the HOHY model, "Cheng et al. (2006)" is the main reference which provides a lot of information listed in the reference of new manuscript (Page 12 Line 5). We have added some necessary and important information of this model, which can be found on Page 7 Line 17-23.

**Q(7)** Page 1 Line 19: Change "access" to "assess"

A: We have changed "access" to "assess" on Page 1 Line 22.

35 Q(8) Page 5 Line 6: "the ehorage". What does it mean?

A: Thanks for pointing out, and we have corrected it. The "Anchorage Ground" is the name of a place; in Chinese it is called "长航锚地". We have modified into "It is called anchorage ground in the unpublished master's thesis with English abstract (Hu, 2006)" which can be found on Page 6 Line 24-25.

**Q(9)** Page 1 Line 5: change "Ministry of Water Resources" to "Ministry of Water Resources of P.R. China"

**A: Thanks for your comments. We have changed "Ministry of Water Resources" to "Ministry of Water Resources of P.R. China", which can be found on Page 1 Line 7.**

5

Q(10) Figure 1(a) and (b): Please show rivers of Wangyu and Huangpu, location of Huangpu Park station in the map. Boundaries of six hydrology units and rivers are not clearly distinguished in the fig. 1. Can they be refined with amendation?

A: Thanks for your comments. Yes, it has been done, please see the new Figure 1.

10

Q(11) Problems with References: the following references were not mentioned in the main text of the manuscript.

(1)Duinker Peter and Greig Lorne: Scenario Analysis in Environmental Impact Assessment: Improving Explorations of the Future, Environmental Impact Assessment Review, 214, 2007

**15 A: Thanks for your comments. We have deleted this reference.**

(2)EA (Environment Agency): Summary of Thames Barrier Flood Defence Closures, Retrieved June 21, 2012, from http:// www.environment-agency.gov.uk, 2012

A: Thanks for your comments. This reference is mentioned in Page 3 Line 3 of our revised manuscript.

(3)JinKe, Wang Chuanhai, Yu Xiaoliang and Lin Hejuan: Application of Joint Regulation Model of 20
 Quantity & Quality in Emergency Water Diversion from Yangtze to Taihu, China Water Resource No.1, 18-20, 2008

A: Thanks for your comments. This reference is mentioned in Figure 2 of our revised manuscript. (4)JinKe: A Methodological Study to Improve Flood Management of the Taihu Lake Basin (Unpublished

25 master's thesis), The International Centre for Water Hazard and Risk Management (ICHARM), Japan, ICHARM Master Paper No.13, 14, 2009

A: Thanks for your comments. This reference is mentioned in Page 7 Line 3 of our revised manuscript.

**Part II Responses to Referee #2**

**General Comments:**

Writing improves a lot in this version and most of my comments are addressed in an appropriate way.However, the response to my general comments on parameter sensitivity and uncertainty is still notconvincing.

**Responses to General Comments:**

A: Thanks for your comments. The simulation results are indeed dependent on the model parameters, and the most sensitive parameters in this study is the roughness coefficient for channels. The smaller

- 10 the roughness coefficient is, the stronger the carrying capacity of the channel is. Meanwhile, uncertainty exists for determining this parameter due to river deposition, channel curvature and so on. In the HOHY model there is a fixed roughness coefficient for each channel segment, which is based on a previous comprehensive river drainage test. Since the year of 1999, the capacity and shape of the TaiPu Canal and Huangpu River have changed little. As a result, the parameters determined
- 15 previously should not have the need to adjust. On the other hand, in this study we focus on the potential impacts of constructing an estuary gate on the flood control in terms of evacuating flood discharge and reducing peak water levels under five different scenarios, all of which are based on the identical HOHY model simulations with the same parameters set. Therefore, we did not put substantial effort in analyzing the sensitivity of parameters in this study.

**Part III Marked-up manuscript version**

Numerical Simulations Analysis of Potential Contribution of the Proposed Huangpu Gate to Flood Control in the Taihu Lake Basinof China

Zhang Hanghui1,2, Liu Shuguang1\*, Ye Jianchun4,2, Pat J.-F. Yeh3

1 Department of Hydraulic Engineering, Tongji University, 200092, Shanghai

2 Taihu Basin Authority of Ministry of Water Resources of P.R.China, 200434, Shanghai

3 Department of Civil and Environmental Engineering, National University of Singapore, 117576, Singapore

\*Correspondence to: Liu Shuguang (liusgliu@tongji.edu.com)

15

5

**Abstract**

The Taihu Lake basin (36895 km2), one of the most developed regions in China located in the hinterland of the Yangtze River Delta, has experienced increasing flood risk. The largest flood in history occurred in 1999 with a return period estimate of 200 years, much higher than the current capacity of flood defense

- 20 with the design return period of 50 years. Due to its flat saucer-like terrain, the capacity of flood control system in this basin depends on flood control infrastructures and peripheral tidal conditions. The Huangpu River, an important river of the basin connecting Taihu Lake upstream and Yangtze River Estuaries downstream, drains two-fifth of the entire basin. Since the water level in Huangpu River is significantly affected by the high tide conditions in estuaries, constructing an estuary gate is considered
- 25 as the most effective solution for flood mitigation. The main objective of this paper is to assess the potential contribution of the proposed Huangpu gate to the flood control capacity of the basin. To achieve this goal, five different scenarios of flooding and the associated gate operations are considered by using numerical model simulations. Results of quantitative analyses show that the Huangpu gate is effective to evacuate floodwaters. It can help to reduce both the peak values and the duration of high water levels in
- 30 the Taihu Lake to benefit the surrounding areas along the Taipu Canal and Huangpu River for more than 100 km2. The contribution of the gate to the flood control capacity is closely associated with its operation modes and duration. For the maximum potential contribution of the gate, the net outflow at the proposed site is increased by 52%. The daily peak level is decreased by a maximum of 0.12m in the Taihu Lake, by a maximum of 0.26-0.37 m and 0.46-0.60m in the Taipu Canal and Huangpu River, respectively, and
- 35 by 0.05-0.39m in surrounding areas along the two rivers depending on local topography. It is concluded that the proposed Huangpu gate can reduce flood risks in the Taihu Lake basin and the surrounding areas along the Taipu Canal and Huangpu River significantly, which is of great benefits to the flood management in the basin and the Yangtze River Delta.

Keywords: Flood control; Huangpu Gate; Taihu Lake Taihu Lake Basin; Numerical analysis

**1** Introduction**

- 5 The Huangpu River, located in the downstream part of the Taihu Lake basin, The Taihu Lake Basin, located in the hinterland of the Yangtze River Delta, Eastern China, is a region which is impacted by both marine, such as tides, waves and the influx of saline water, and river, such as flows of fresh water and siltation. Extensive urban development has contributed to increasing flood magnitude and flood frequency in this region. Major flood disasters, flooding more than 3000 km2, have occurred more than
- 10 ten times during the twentieth century (Yu et al. 2000). The largest flood disaster occurred in 1999 and it resulted in damages with direct economic loss of 16 billion USD (Wang et al., 2011). There are 239 typhoons affecting the basin from 1949 to 2013, averaging about 3 to 4 per year on average (Ye & Zhang, 2015). According to the report from The Intergovernmental Panel on Climate Change (IPCC), the flood control of coastal systems and low lying areas is addressed, including the Yangtze River delta, which is
- 15 identified as one of the highly vulnerable coastal deltas (2007). The Huangpu River\_is the main shipping and drainage route to the port city Shanghai in China. The state of the Taihu Lake Taihu Lake basin, is the last significant tributary of the Vangtze River. The Huangpu River is not only the major river that drains the local floodwater of the Shanghai city but also one of the major rivers that drains the floodwater of the Taihu Lake Basin. With
- 20 a length of 113 km, it flows through the urban core of Shanghai city, which is evaluated as one of the most vulnerable metropolises to extreme flooding in the world (Balica et al., 2012). Wang et al. (2012) Another research predicted that half of Shanghai will be flooded and 46% of seawalls and levees will be overtopped in 2100 (Wang et al. 2012), causing serious urban flooding. Typhoon is one of main natur<del>rual</del> factors to trigger flood disaster in this area. When typhoon is comingcomes, the storm surges caused will
- be driven into the Yangtze River estuary to further -causingincrease storm tide levelsto-increase additionally because of due to the shallow waters and confined dimensions within the estuary (Nai et al., 2004). When this conjunctscoincides with the astronomical high tides, the storm tide traveling into the Huangpu River can easirapidly raise water levels in river and possibley cause inundation of the urban areas of Shanghai. It has been reported that along with global climate change, the frequency and intensity of typhoons have increased substantially (Qin et al., 2005).
  - Taihu Lake Taihu Lake is located about 80 kilometers away west of the Shanghai city center (Figure 1). The Huangpu River is not only the major river draining local-floodwaters of both the Shanghai city but alsoand Taihu Lake basin the major routine that drains the floodwater from the lake. After the completion of eleven key projects for the integrated water resources management in Taihu lakeTaihu
- 35 Lakethe basin, the discharge from the upper reach of Huangpu River increased correspondingly, which resultsing in a considerable rise of water level rise in the Huangpu River (Zhou et al., 2016). The river embankments, which is an traditional flood defense infrastructure, has been was built along the Huangpu River since in the 1950s. Its flood control capacity, however, is 
[revised manuscript text omitted]
 hittaffecting the basin during thefrom 1949-to-2013 period, averaging on average about 3 to 4 per year on average (Ye & and Zhang, 2015). According to the recent assessment report (AR5)

compiled byfrom tThe Intergovernmental Panel on Climate Change (IPCC, 2013) where the flood control of coastal systems and low-lying areas is addressed, the flood control of coastal systems and low lying areas is addressed, including the Yangtze River deltadelta, which is identified as one of the highly vulnerable coastal deltas in the world-(2007).

Generally, the basin is characterized by thea monsoonal climate with the period concentrated in summer (,-from June to July), lasting several weeks or even months. Consequently, thebroad\_-scale 30 rainfall events occur frequently are prone to occurring with andue to excessive magnitude arainfnd a all with long duration, which contributinges to the basin-wide floods. Meanwhile, the monsoon flood risks are exacerbated by the very low lying topography and high tide conditions of peripheral outlets in the basin. The\_largest flood in history occurred in 1999, and the total rainfall in a the 43-days monsoon period reached 670 mm, which was three times more than the long-term (1954-2010) average

- 35 precipitation\_during the same period of time of the year in the 1954-2010 periods. The return period of 1999 flood event is estimated as 200-years (Evans and& Cheng, 2010), much highlarger than both the current-capacity of 50-year design return period of the flood control capacity of in the basin with 50 years return period and the pprojected\_lanned capacity with 100 year\_designs return period in 2025 (MWR, 2008). MTotal\_ean totalsaverage rainfall of the 7-day, 15-day, 30-day, 45-day, 60-day and 90-day
- 40 accumulated rainfall in 1999 all exceeded the historical records values recorded (Wu, 2000). During this

flood, the high water level in Taihu Lake Taihu Lake set a new record of 5.08 m, which exceedinged the previous record set in the 1991 flooddesign water level of 50-year return period by 0.29-43 m.

In the basin, there are numerous tidal channels-that linking-the lakes and the coast (bay, estuary), and most outlets are controlled by the floodgates subject to tidal locking tide locking. The Huangpu River

- 5 meandering s-through the downtown area of Shanghai City -and-connects the westward-located Taihu Lake with the Yangtze River estuary in the North East, as shown in Fig. 1(a). The Huangpu River is 113 kilometerskilometers long, with a depth of 5-15 meters, and a width of 300-500 meters (800 m at the estuary), formed by - It is born from the convergence of Xietang riverRiver originated from the Taihu Lake and Area Area Yangchengdianmao, Yuanxiejing Creek and DamMaogang Creek from Area
- 10 originated from Area Hangjiahu. The river then-flows proceeds through the downtown Shanghai before i and finally injectflowings into the YangzteYangtze River at the estuary mouth of Wusong, -mouth as the only one without an estuary gate in the basin. The tidal effect complicates the flow patterns of the Huangpu River, and helps to keep the floodwater in the river. Generally, the river can naturally drain floodwaters only for 13-14 hours per day. The Huangpu River, as the only one without an estuary gate,
- 15 flows into the Yangtze River estuary and experiences two high tides and two low tides each day (semi-diurnal tides), and receivinges about 40.9 billion m3 of tidal water from the Yangtze River (Zhang, 1997)., as shown in Fig. 1(a). For this reason, tThe tidal effect complicates the flow patterns of the Huangpu River. As the 'barrier effect', the high tide helps to keep the floodwater in the river. Generally, Tthe river can, therefore, naturally drain floodwaters in the Taihu Lake and the middle of the floodplain only
- 20 for 13-14 hours per day. The Huangpu River receives about 40.9 billion m3 of tidal water from the Yangtze River (Zhang, 1997). The total tidal influx of the Huangpu River is about 47.47 million m3-/per year, and the total inflow from its upstream area is about 10.02 billion m3/-per year. Its sediment concentration upstream is of 0.049 kg/m3, and that is of 0.213 kg/m3 downstream (Yan, 1992). The problem caused by sediment and siltation is not serious in this river because the inflow from the upstream

25 area is far more than the tidal water from the estuary.

**3 Methodology**

**3.1 Description of five scenarios**

It is instructive to investigate the potential contribution of the proposed Huangpu gate to the flood control of the Taihu Lake basin, which is still in the preliminary demonstration-of-benefit stage currently.

30 The main research strategy used in this study is the scenario analysis\_based on numerical model simulations, which is a process of analysing possible future events by considering alternative possible outcomes.

It is instructive to investigate the potential contributions of the proposed Huangpu gate to the flood control of the basin, which is still in the preliminary demonstration-of-benefit stage.

35 In total five different scenarios are are-considered used in this study as summarized in Table 1. Among them, the firstone scenario considers the case is the case-without gate construction, and all the remaining f. Four scenarios considerare the cases with with proposed gate construction, but each with different operational rules for the comparison to others. TSpecific ally speaking, s'Scenario "base A" is used as the

basis for comparison with other scenarios (Table 1). It- represents the case-means that the estuary gate is would-not be-constructed at the Huangpu River mouth, which is used as a basis for comparison. The s-Scenarios "A1" - and "-Scenario-A2" - are designed for the quantitative analysis\_of tanalyze-heits potential benefits due to gate operation in draining the-floodwaters from the Taihu L-lake and its-the

5 upstream areas. The s'Scenario "A3" is designed to quantitatively-analyze theits contribution (function) of gate construction to block tide intrusion. The last sscenario, "A4", is the-just a case for to-analyzinge the potential maximum benefits to flood control of the basin. TAll he numerical simulations for the five different the scenarios are all based on the conditions during the 1999 flood event, which is, the largest flood in history for the study area.

10

40

For the s'Scenario "base A",  $\stackrel{,}{\rightarrow}$  the estuary gate iswould not be constructed at the outlet of the Huangpu River. Thus, the water in the Huangpu River and the Yangtze River estuary can exchange naturally.

For the scScenario "A1"2 the proposed gate will be operated in the rising stage of the lake levels according which is to create a new record with a high possibility to create new record of the lake in level based onseveral days due to weather forecast. In the model simulation of the 1999 flood event, it began to operate seven days in advance before the lake level reached its peak value.

For the s4Scenario "A2", 2 the proposed gate will be operated when the large basin-wide floods
 occur with the lake level higher than 4.50 m, which meanings a severe and urgent flooding -situation inof
 the Taihu LakeTaihu Lake which requires is very severe anthe acceleration of floodwater drainage -of
 the major downstream rivers, including the Huangpu River.

- For the scScenario "A3", a2: portionart of the tidal water intrusion would ill be blocked, by the gate, and the gate will remain opennot be closed to prevent tide intrusion until the tide rises to a threshold (defined in this study as 4.0m). That is, the gate will not be closed for blocking tide intrusion every day; instead, it will only be closed underfor the situationcase when the high water exceeds the tide threshold and is forecasted to continue rise.
- 30 For the last s-Scenario "A4", -: the gate would prevent all the intrusion of tidal water intrusion during the flooding period, representing a hypothetical extreme case since it is not practical in implementation owing to the difficulty inof frequent operation (e.g., i.e. to close twice every day) of such a huge gate with a width of about 400—500 m. This scenario is merelyjust a case for analyzing the potential maximum benefits to flood control of the basin. Indeed, Furthermore, it is not necessary to blockprevent all tidal
- 35 water intrusionintrusion, which would, entirely resulting in a stop any of water exchange between the Huangpu River and the Yangtze River networks. Under suchthis condition, it is also likely to produce a negative impacts on both waterway transportation and water environment system.

Considering the time needed for making policy decision and gate construction, it is highly likely that the Huangpu gate will not be completed and start to operate work after in 2025. For this reason, the proposed flood control projects in the plan designed by in the Plan (MWR (, 2008) will also be

incorporated as the are also simulated in the scenarios of aflood defense infrastructure in this studys. TAlso, he model parameters inof the numerical simulations model in this study are specified as the same as those used in the design plan by in the Plan (MWR (, 2008). Hu (2006) proposed the Aanchorage Geround located at the mouth of Huangpu River (as shown in Figure 1) as the best

- 5 site for the gate construction since, since it is in the first regular bank and hence theits negative impacts to the shipping and navigation are the least due to its location. Cui (2012) and Lu (2008) proposed the sameimilar location for gate construction. Similarly for In the numerical model simulations in of this study, the estuary gate will also be constructed simulated at the Aanchorageehorage Gground, the place name in the unpublished master's thesis with English abstract (Hu, 2006), which is, place which is about
- 10 5-6 km from the Huangpu Rriver outletmouth.

**3.2 Model description**

The HOHY model, developed by the Hohai University\_in\_, China; is used in this study. This model has been was tested for numerous regional applications since 1970s, and was applied to study the whole Taihu Lake basin in 1997[YJ1]. It is one of the main productions of a three-year water study atim the Taihu
Lake basin\_supported by the World Bank and, which were-jointly undertaken by the Hohai University and the Delft Hydraulics in, the Netherlands. The HOHY model can simulate the cycle of flood-waters well. Meanwhile, the model can provide sa broad-scale, at a broad scale, a reasonable simulation of the flooding control system in the Taihu Lake Taihu Lake basin-flooding system. It can simulate not only simulate complex hydro-systems with numerous interlaced rivers and lakes, complicated relationships
between river nets, hilly topography and tidal boundaries, and also the simulate complex operational rules of control structures, such as sluices, pumps and siphons. This model htt as been has been utilized inused in a variety of past studiesfields, such as the preliminary demonstration-of-benefit stage of water works in the Taihu Lake basin. In particular, the model has been successfully applied in the flood control planning of the Taihu Lake basin as, which was approved by the Minister of Water Resources

25 of P.R.C in 2008 WMR (MWR, 2008).

The model is composed of two parts: a hydrological part for simulating runoff generation and routing and a hydraulic part for simulating channel flows. Each of them can run independently. The schematization of the model is shown in Fig. 2. More details of the model can be found in Cheng et al. (2006) and Jin (2009)

30 (2006) and Jin (2009).

Runoff is generated when precipitation exceeds the infiltration, interception and depression storage. The basin land use is classified into four types: water surface, paddy field, non-irrigated farmland and constructed land. Each of them employs different parameterizations to calculate runoff generation. Runoff is then routed according to basin topography. In hilly areas, the instantaneous unit hydrograph

35 method is used, considering the store and drainage processes of reservoirs and large ponds. In plain areas, the method of runoff curve number is used for each computed area.

After the runoff from the hilly and plain areas flows into river networks, the hydraulic method is applied for simulation of river flow. Only those lakes with larger water surface are considered as possessing the function of storing floodwater, while the others are considered as intersections like the links among rivers. The operation of water-engineering works such as the simulations of gates, pumping stations and siphons will be simulated in the model. The Saint Venant Equations are used as the governing equations for the 1-D unsteady open channel flow, including the continuity\_equation (1) and momentum equations (2) as follows

5

10

$$\frac{\partial Q}{\partial x} + \frac{\partial A}{\partial t} = q_L \tag{1}$$

$$\frac{\partial Q}{\partial t} + \frac{\partial}{\partial x} \left(\alpha \frac{Q^2}{A}\right) + gA \frac{\partial Z}{\partial x} + gA \frac{n^2 |Q| Q}{R^{1.333}} = q_L v_x$$
(2)

[revised manuscript text omitted]

- Wang, LiangL., -Cai, YongliY. L., Chen, HongquanH. Q., Dag, Daler, Zhao, JingminJ. M., and Yang, JuanJ.: Flood disaster in Taihu Basin, China: casual chain and policy\_option analyses, Environ Earth Sci., doi DOI 10.1007/s12665-010-0786-x, 1119-1124, 2011.
- Wu, TailaiT. L.: 1999 Catastrophic Flood in Taihu Basin and the Consideration for Taihu Flood Control
- Planning, Journal of, Lake Sciences, Vol.12, No.1, 6-11, 2000 (in Chinese with English abstract). Xiao, Y., Feng, W., and Zhang, Y. Q.: Benefits of building estuarine tidal gates in China and other countries and its impact analysis, China Water Resource, 25-28, 2017.

Yao, WeichengW. C.: The study on the eEngineering dDesign of eEstuary gGate at the mMouth of the Huangpu River, Shanghai Water, 21-24, 2001-(in Chinese).

- 10 Yan, Kai.: China coastal engineering, Hohai University Press, 1992-(in Chinese).
  - Ye, Jianchun-J. C., and Zhang, Hanghui H. H.: Practices and thinking of flood risk management in Taihu Lake Taihu Lake Basin, Advances in Science and Technology of Water Resources, Vol. 35, No. 5, DOI:10. 3880/ j. issn. 1006 7647, 2015 (in Chinese with English abstract).
  - Yua X.-Ga, Wua T.-La, and Jianga J.-Ha: 1999 flooding Heavy in the Taihu\_Basin: investigation, analysis and future suggestions on the integrated harnessing in the basin. J Lake Sci 12(1):1–5, 2000-(in Chinese with English abstract).
  - Zhang, ChonghuaC. H.: Case Study II\* Shanghai Huangpu River, Water Pollution Control A Guide to the Use of Water Quality Management Principles Edited by Richard Helmer and Ivanildo Hespanhol, Published on behalf of the United Nations Environment Programme, the Water Supply
- 20 & Sanitation Collaborative Council and the World Health Organization by E. & F. Spon, ISBN 0 419 22910 8, 1997
  - Zhou, Z.Z.-zhengzheng, Liu, ShuguangS.G., Zhong, GuihuiG.H., and Cai, Y.i: Flood Disaster and Flood Control Measurements in Shanghai. Natural Hazards, ISSN 1527-6988, 2016.

25

15